# Respiratory alkalosis provokes spike-wave discharges in seizure-prone rats

Kathryn A Salvati[1,2]\*, George MPR Souza[1], Adam C Lu[1,2], Matthew L Ritger[1,2], Patrice Guyenet[1], Stephen B Abbott[1], Mark P Beenhakker[1]\*

[1]Department of Pharmacology, University of Virginia, Charlottesville, United States; [2]Neuroscience Graduate Program, University of Virginia, Charlottesville, United States

**Abstract** Hyperventilation reliably provokes seizures in patients diagnosed with absence epilepsy. Despite this predictable patient response, the mechanisms that enable hyperventilation to powerfully activate absence seizure-generating circuits remain entirely unknown. By utilizing gas exchange manipulations and optogenetics in the WAG/Rij rat, an established rodent model of absence epilepsy, we demonstrate that absence seizures are highly sensitive to arterial carbon dioxide, suggesting that seizure-generating circuits are sensitive to pH. Moreover, hyperventilation consistently activated neurons within the intralaminar nuclei of the thalamus, a structure implicated in seizure generation. We show that intralaminar thalamus also contains pH-sensitive neurons. Collectively, these observations suggest that hyperventilation activates pH-sensitive neurons of the intralaminar nuclei to provoke absence seizures.

## Editor's evaluation

The study evaluates the long debated question of how respiration affects seizure susceptibility. The authors use a rigorous approach to manipulate the gases breathed in by seizure prone rats while monitoring their respiration, electroencephalographic activity, blood pH and gas levels. They show that changes in pH caused by hyperventilation drive spike-wave seizures, that optogenetically driving hyperventilation induced spike-wave seizures by changing pH, and that intralaminar nuclei in the thalamus contain neurons that are activated during hyperventilation and are pH sensitive.

\*For correspondence:
kathryn.salvati@ucsf.edu (KAS);
markbeen@virginia.edu (MPB)

**Competing interest:** The authors declare that no competing interests exist.

## Introduction

Epilepsy is a common neurological disorder characterized by recurrent and spontaneous seizures. Yet, accumulating evidence indicates that seizures are not necessarily unpredictable events (*Amengual-Gual et al., 2019*; *Bartolini and Sander, 2019*; *Baud et al., 2018*; *Ferlisi and Shorvon, 2014*). Several factors affect seizure occurrence, including metabolism (*Lusardi et al., 2015*; *Masino et al., 2012*; *Masino and Rho, 2012*; *Masino and Rho, 2019*), sleep (*Bazil, 2019*; *Fountain et al., 1998*; *Malow et al., 1999*; *Nobili et al., 2001*), catamenia (*Herzog et al., 2014*; *Joshi and Kapur, 2019*; *Reddy et al., 2001*), light (*Padmanaban et al., 2019*), and circadian rhythm (*Amengual-Gual et al., 2019*; *Debski et al., 2020*; *Smyk and van Luijtelaar, 2020*; *Stirling et al., 2021*). In extreme cases, stimuli immediately provoke seizures, a condition known as *reflex epilepsy* (*Kasteleijn-Nolst Trenité, 2012*; *Koepp et al., 2016*). The mechanisms that render certain seizure-generating networks susceptible to external factors remain unknown.

A highly reliable seizure trigger associated with childhood absence epilepsy is hyperventilation. Between 87% and 100% of all children diagnosed with the common *genetic generalized epilepsy* produce spike-wave seizures upon voluntary hyperventilation (*Hughes, 2009*; *Ma et al., 2011*; *Sadleir*

*et al., 2009*). Indeed, hyperventilation serves as a powerful tool for diagnosing this childhood epilepsy (*Adams and Lueders, 1981*; *Holowach et al., 1962*; *Sadleir et al., 2006*; *Watemberg et al., 2015*). Remarkably, as no single genetic etiology drives absence epilepsy (*Chen et al., 2013*; *Crunelli and Leresche, 2002*; *Helbig, 2015*; *Koeleman, 2018*; *Robinson et al., 2002*; *Xie et al., 2019*), hyperventilation appears to recruit fundamental seizure-generating mechanisms shared virtually by all patients.

Exhalation of $CO_2$ during hyperventilation causes hypocapnia, a state of decreased arterial $CO_2$ partial pressure ($PaCO_2$), and respiratory alkalosis, a state of elevated arterial pH (*Laffey and Kavanagh, 2002*). Hyperventilation also causes rapid arterial vasoconstriction (*Raichle and Plum, 1972*) and increased cardiac output (*Donevan et al., 1962*). Recent work demonstrates that inspiration of 5% $CO_2$ blunts hyperventilation-provoked spike-wave seizures in humans (*Yang et al., 2014*). Collectively, these observations suggest that respiratory alkalosis serves as the primary trigger for hyperventilation-provoked absence seizures.

Spike-wave seizures associated with absence epilepsy arise from hypersynchronous neural activity patterns within interconnected circuits between the thalamus and the cortex (*Avoli, 2012*; *Beenhakker and Huguenard, 2009*; *Huguenard and McCormick, 2007*; *McCafferty et al., 2018*; *McCormick and Contreras, 2001*; *Meeren et al., 2002*). The crux of the prevailing model describing absence seizure generation includes an initiating bout of synchronous activity within the somatosensory cortex that recruits rhythmically active circuits in the thalamus (*Meeren et al., 2002*; *Sarrigiannis et al., 2018*). With widespread connectivity to the cortex, the thalamus then rapidly generalizes spike-wave seizures to other brain structures. The extent to which thalamocortical circuits respond to shifts in pH during hyperventilation-induced respiratory alkalosis is unknown.

Herein, we test the hypothesis that respiratory alkalosis regulates the occurrence of spike-wave seizures. We demonstrate that hyperventilation-provoked absence seizures observed in humans can be mimicked in an established rodent model, the WAG/Rij rat (*Coenen and van Luijtelaar, 2003*; *Coenen et al., 1992*; *Russo et al., 2016*; *van Luijtelaar and Coenen, 1986*). We first show that hyperventilation induced with hypoxia reliably evokes respiratory alkalosis and increases spike-wave seizure count in the WAG/Rij rat. When supplemented with 5% $CO_2$ to offset respiratory alkalosis, hypoxia did not increase spike-wave seizure count. Moreover, hypercapnia alone (high $PaCO_2$) reduced spike-wave seizure count despite a robust increase in respiration rate. We also show that optogenetic stimulation of brainstem respiratory centers to produce respiratory alkalosis during normoxia induces $CO_2$-sensitive spike-wave seizures. Collectively, these results identify respiratory alkalosis as the primary seizure trigger in absence epilepsy following hyperventilation. Finally, we show that structures of the intralaminar thalamic nuclei are both (1) activated during respiratory alkalosis and (2) pH sensitive. Thus, our data demonstrate that respiratory alkalosis provokes spike-wave seizures and shine a spotlight on the poorly understood intralaminar thalamus in the pathophysiology of spike-wave seizures.

## Results

### Hypoxia triggers spike-wave seizures in the WAG/Rij rat

We first set out to determine if an accepted rat model of absence epilepsy, the WAG/Rij rat, recapitulates hyperventilation-provoked absence seizures, as observed in humans. We combined whole-body plethysmography and electrocorticography/electromyography (ECoG/EMG) recordings in awake WAG/Rij rats to assess respiration and spike-wave seizure occurrence while exposing animals to different gas mixtures of $O_2$, $CO_2$, and $N_2$ (*Figure 1A and B*). We only considered spike-wave seizures that persisted for a minimum of 2 s and occurred concomitantly with behavioral arrest in the animal. Spike-wave seizures are distinguishable from non-REM sleep based on the appearance of 5–8 Hz frequency harmonics in the power spectrogram (see *Figure 1B*, *expanded trace*).

We first compared respiration and ECoG/EMG activity in rats exposed to atmospheric conditions (i.e. normoxia: 21% $O_2$; 0% $CO_2$; 79% $N_2$) and hypoxia (10% $O_2$; 0% $CO_2$; 90% $N_2$). Hypoxia reliably stimulates rapid breathing, blood alkalosis, and hypocapnia in rats (*Basting et al., 2015*; *Souza et al., 2019*). We cycled rats between 40 min epochs of normoxia and 20 min epochs of hypoxia. $O_2$ levels were measured from the outflow of the plethysmography chamber for confirmation of gas exchange (*Figure 1B*, *top*). Hypoxia evoked a robust increase in respiratory rate (*Figure 1B*, *expanded*) and reliably provoked seizures. A peristimulus time histogram (PSTH) aligned to the onset of gas exchange shows spike-wave seizure counts during the 15 min immediately before and during hypoxia

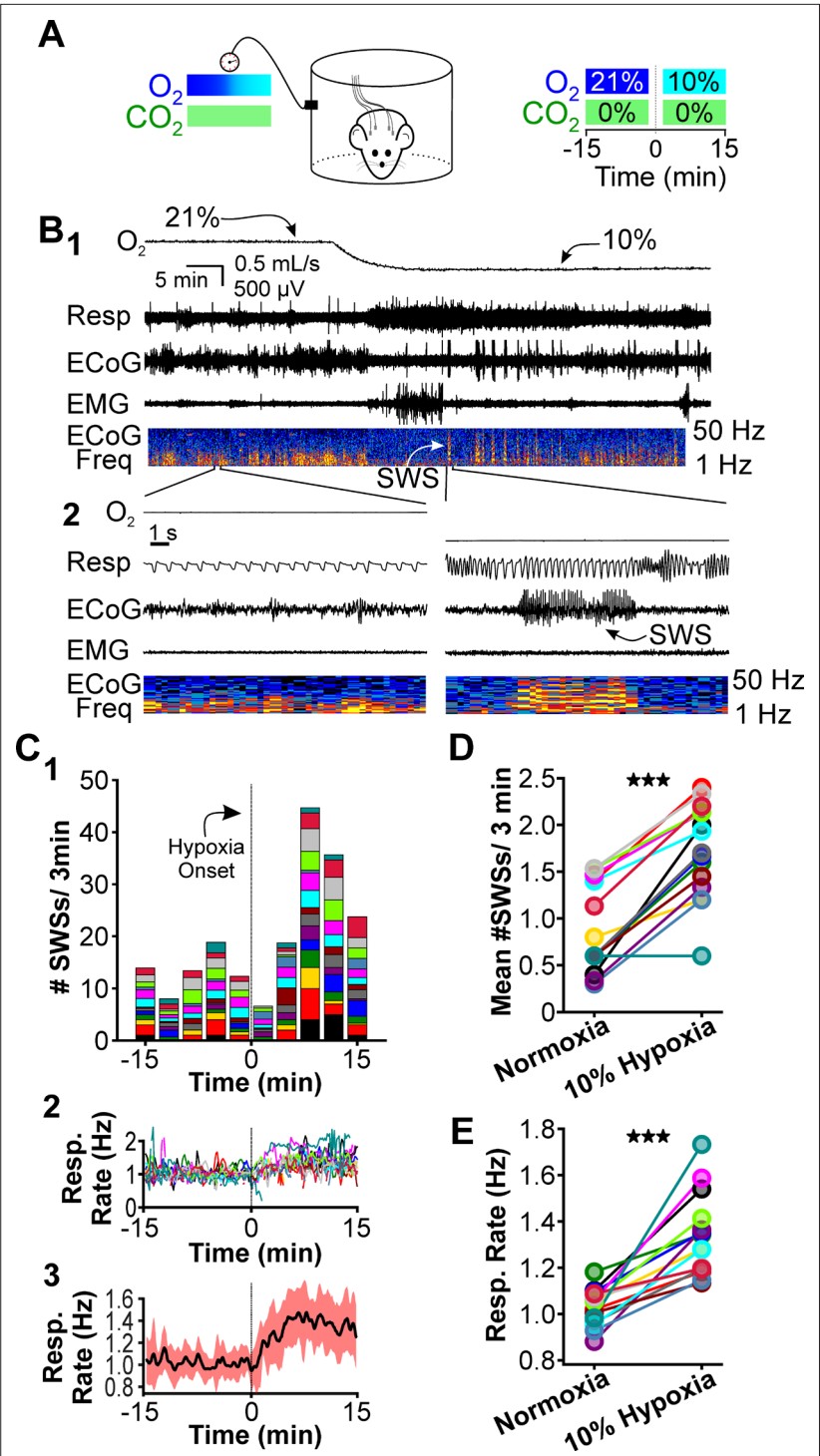

**Figure 1.** Hypoxia provokes hyperventilation-associated spike-wave seizures (SWS) in WAG/Rij rats. (**A**) Experimental approach. *Left:* plethysmography chambers recorded ventilation and electrocorticography/electromyography (ECoG/EMG) signals in rats exposed to normoxia (i.e. 21% $O_2$) and hypoxia (i.e. 10% $O_2$). *Right:* example gas exchange protocol used to generate the peristimulus time histogram in panel C. SWS count was measured during the 15 min before and after gas exchange at t = 0 min. (**B**) Representative recordings during transition from normoxia to hypoxia. (1) From top to bottom: chamber $O_2$, respiration, ECoG, EMG, and ECoG power spectrogram. White arrow points to SWS. (2) *Bottom:* expanded view B1. Spectrogram reveals 5–8 Hz frequency harmonics associated with SWS. (**C**) SWS and respiration quantification. (1) Stacked histogram illustrating SWS count for each animal before and after the onset of hypoxia; each color is a different rat. Arrow points to gas

*Figure 1 continued on next page*

*Figure 1 continued*

exchange at t = 0 min. (2) Corresponding respiratory rate for each animal shown in panel C1. (3) Mean respiratory rate for all animals. (**D**) Mean SWS count per bin and (**E**) respiratory rate before and after gas exchange. See for *Tables 1–4* detailed statistics. ***p < 0.001.

(*Figure 1C1*); the PSTH shows the contribution of each rat in stacked histogram format. Respiratory rates confirmed that hypoxia increased ventilation (*Figure 1C2,3*). To quantify the effect of hypoxia on seizures, we calculated the mean spike-wave seizure count across all bins for each rat. Relative to normoxia, spike-wave seizure count during hypoxia was nearly twofold higher (p = $4.5 \times 10^{-7}$, n = 15; *Figure 1D*, *Table 1*) and respiratory rate increased by 30% (p = $1.6 \times 10^{-5}$, n = 15; *Figure 1E*, *Table 2*). Whereas the duration of individual spike-wave seizures was not altered by hypoxia (normoxia: 5.3 ± 0.4 s; hypoxia: 5.8 ± 0.4 s; p = 0.56, n = 15, *Table 3*), the frequency of individual events was lower (normoxia: 7.6 ± 0.12 Hz; hypoxia: 5.8 ± 0.4 Hz; p = $4.7 \times 10^{-5}$, n = 15, *Table 4*).

Recent work shows that spike-wave seizures commonly occur in several rat strains, including those that are generally not considered epileptic (*Taylor et al., 2017*; *Taylor et al., 2019*). While between 62% (*Vergnes et al., 1982*) and 84% (*Robinson and Gilmore, 1980*) of Wistar rats do not have seizures, we nonetheless tested whether hypoxia can unmask seizure-generating potential in this strain, as Wistar and WAG/Rij rats share the same genetic background (*Festing, 1979*). In normoxia, seizures were absent in all four Wistar rats we tested, consistent with the infrequent spike-wave seizure occurrence reported for this strain. Relative to normoxia in Wistar rats, hypoxia induced hyperventilation, hypocapnia, and blood alkalization but did not provoke spike-wave seizures (*Figure 2*; see *Table 5*). Instead, hypoxia primarily triggered arousal in Wistar rats, as revealed in EEG spectrograms by the reduction in sleep-related frequencies. Therefore, we hypothesize that hypoxia-provoked spike-wave seizures are unique to seizure-prone rodent models, just as hyperventilation does not provoke absence seizures in otherwise healthy humans.

## $CO_2$ suppresses spike-wave seizures

Hyperventilation promotes hypocapnia, a state of low $PaCO_2$. As dissolved $CO_2$ is acidic, hyperventilation-triggered hypocapnia is also associated with respiratory alkalosis. To test the hypothesis that hypocapnia specifically provokes seizures, we next determined whether supplemental $CO_2$ (5%) blunts the spike-wave seizure-provoking effects of hypoxia. We performed ECoG/plethysmography experiments as before but alternated between two test trials: hypoxia and hypoxia/hypercapnia (10% $O_2$, 5% $CO_2$; 85% $N_2$). Test trials were interleaved with 40 min periods of normoxia to allow blood gases to return to baseline levels (*Figure 3A*). As before, hypoxia increased spike-wave seizure count

**Table 1.** Spike-wave seizure count.

| Figure | Comparison | Bin count (mean ± S.E.) | n | p Value |
|---|---|---|---|---|
| | Normoxia | 0.89 ± 0.12 | | |
| 1D | Hypoxia | 1.73 ± 0.13 | 15 | $4.5 \times 10^{-7}$ |
| | Normoxia | 0.99 ± 0.18 | | |
| 3C | Hypoxia | 1.82 ± 0.14 | 9 | $1.76 \times 10^{-6}$ |
| | Normoxia | 1.09 ± 0.22 | | |
| 3F | Hypoxia + $CO_2$ | 0.84 ± 0.13 | 9 | 0.18 |
| | Normoxia | 1.36 ± 0.17 | | |
| 4C | Normoxia + $CO_2$ | 0.95 ± 0.10 | 8 | 0.0028 |
| | Normoxia | 1.17 ± 0.38 | | |
| 5D | Normoxia + Photostim. | 2.27 ± 0.63 | 10 | 0.002 |
| | Normoxia | 1.04 ± 0.32 | | |
| 5G | Normoxia + Photostim.+ $CO_2$ | 1.01 ± 0.30 | 6 | 0.86 |

**Table 2.** Respiratory rate.

| Figure | Comparison | Resp. rate (Hz) (mean ± S.E.) | n | p Value |
|---|---|---|---|---|
| | Normoxia | 1.03 ± 0.02 | | |
| 1E | Hypoxia | 1.33 ± 0.05 | 15 | $1.67 \times 10^{-5}$ |
| | Normoxia | 1.00 ± 0.02 | | |
| 3D | Hypoxia | 1.28 ± 0.05 | 9 | $6.59 \times 10^{-4}$ |
| | Normoxia | 1.06 ± 0.03 | | |
| 3G | Hypoxia + $CO_2$ | 1.88 ± 0.15 | 9 | $2.71 \times 10^{-4}$ |
| | Normoxia | 0.99 ± 0.03 | | |
| 4D | Normoxia + $CO_2$ | 1.78 ± 0.10 | 9 | $3.78 \times 10^{-5}$ |
| | Normoxia | 1.02 ± 0.03 | | |
| 5E | Normoxia + Photostim. | 1.24 ± 0.08 | 10 | 0.019 |
| | Normoxia | 1.01 ± 0.03 | | |
| 5H | Normoxia + Photostim.+ $CO_2$ | 1.84 ± 0.08 | 6 | 0.031 |

by nearly twofold (p = $1.76 \times 10^{-6}$, n = 9; *Figure 3B1 and C*, *Table 1*) and increased respiratory rate by 27% (p = $6.59 \times 10^{-4}$, n = 9; *Figure 3B3 and D*, *Table 2*). Also as before, the duration of individual spike-wave seizures was not altered by hypoxia (normoxia: 5.5 ± 0.5 s; hypoxia: 6.3 ± 0.5 s; p = 0.26, n = 9, *Table 3*), but the frequency of individual events was lower (normoxia: 7.7 ± 0.2 Hz; hypoxia: 6.3 ± 0.5 Hz; p = 0.014, n = 9, *Table 4*). In the same rats, supplementing hypoxia with 5% $CO_2$ suppressed the spike-wave seizure response insofar that hypoxia/hypercapnia did not change spike-wave seizure count relative to normoxia (p = 0.18, n = 9; *Figure 3E1 and F*, *Table 1*) despite a predictable and robust elevation in respiratory rate (p = $2.71 \times 10^{-4}$, n = 9; *Figure 3E2, and G*, *Table 2*). Relative to normoxia, the duration of individual spike-wave seizures was elevated during hypoxia/hypercapnia (normoxia: 5.5 ± 0.6 s; hypoxia/hypercapnia: 7.5 ± 1.0 s; p = 0.006, n = 9), but the frequency of individual spike-wave seizures was unchanged (normoxia: 7.8 ± 0.1 Hz; hypoxia/hypercapnia: 7.5 ± 1.0 Hz; p = 0.77, n = 9, *Table 4*).

In a separate cohort of rats, we collected arterial blood samples to measure blood $PaCO_2$, $PaO_2$, and pH during normoxia, hypoxia, and hypoxia/hypercapnia (see *Table 6*). We observed a considerable

**Table 3.** Spike-wave seizure duration.

| Figure | Comparison | Duration (s) (mean ± S.E.) | n | p Value |
|---|---|---|---|---|
| | Normoxia | 5.3 ± 0.4 | | |
| 1D | Hypoxia | 5.8 ± 0.4 | 15 | 0.56 |
| | Normoxia | 5.5 ± 0.5 | | |
| 3C | Hypoxia | 6.3 ± 0.5 | 9 | 0.26 |
| | Normoxia | 5.5 ± 0.6 | | |
| 3F | Hypoxia + $CO_2$ | 7.5 ± 1.0 | 9 | 0.006 |
| | Normoxia | 5.6 ± 0.4 | | |
| 4C | Normoxia + $CO_2$ | 6.2 ± 0.4 | 8 | 0.22 |
| | Normoxia | 4.3 ± 0.6 | | |
| 5D | Normoxia + Photostim. | 4.6 ± 0.5 | 10 | 0.51 |
| | Normoxia | 6.8 ± 1.0 | | |
| 5G | Normoxia + Photostim.+ $CO_2$ | 6.9 ± 1.0 | 6 | 0.88 |

**Table 4.** Spike-wave seizure frequency.

| Figure | Comparison | Frequency (Hz) (mean ± S.E.) | n | p Value |
|---|---|---|---|---|
| 1D | Normoxia | 7.6 ± 0.1 | | |
| | Hypoxia | 5.8 ± 0.4 | 15 | $4.7 \times 10^{-5}$ |
| 3C | Normoxia | 7.7 ± 0.2 | | |
| | Hypoxia | 6.3 ± 0.5 | 9 | 0.014 |
| 3F | Normoxia | 7.8 ± 0.1 | | |
| | Hypoxia + $CO_2$ | 7.5 ± 1.0 | 9 | 0.18 |
| 4C | Normoxia | 6.1 ± 1.0 | | |
| | Normoxia + $CO_2$ | 6.2 ± 0.0 | 8 | 0.28 |
| 5D | Normoxia | 7.5 ± 0.2 | | |
| | Normoxia + Photostim. | 4.6 ± 0.5 | 10 | $2.2 \times 10^{-4}$ |
| 5G | Normoxia | 7.9 ± 0.2 | | |
| | Normoxia + Photostim.+ $CO_2$ | 6.9 ± 1.0 | 6 | 0.33 |

change in $PaO_2$ [F (1.056, 5.281) = 406.4, p = $3.0 \times 10^{-6}$], $PaCO_2$ [F (1.641, 8.203) = 338.9, p = 1.9 × $10^{-8}$] and pH [F (1.938, 9.688) = 606, p = $7.2 \times 10^{-11}$] values among the three conditions. Hypoxia decreased $PaCO_2$ (p = $2.1 \times 10^{-6}$ n = 6; **Figure 3H2**, **Table 6**) and concomitantly alkalized the blood (p = $7.0 \times 10^{-6}$, n = 6; **Figure 3H3**, **Table 6**). We also observed a decrease in $PaO_2$ (p = $6.0 \times 10^{-6}$, n = 6; **Figure 3H1**, **Table 6**). Supplemental $CO_2$ returned blood pH (p = 0.008, n = 6; **Figure 3H3**, **Table 6**) and $PaCO_2$ (p = 0.42, n = 6; **Figure 3H2**, **Table 6**) to normoxia levels. However, heightened respiratory rate in supplemental $CO_2$ raised $PaO_2$ (p = 00013, n = 6; **Figure 3H1**, **Table 6**). Collectively, these data support the hypothesis that blood pH powerfully regulates spike-wave seizure activity.

Next, we tested whether supplementing normoxia with 5% $CO_2$ is sufficient to reduce spike-wave seizure counts. Respiration during high $CO_2$ causes hypercapnia, a condition that increases blood $PaCO_2$ and acidifies the blood (**Eldridge et al., 1984**). As with hypoxia, hypercapnia also triggers hyperventilation (**Guyenet et al., 2019**). We performed ECoG/plethysmography experiments in rats that cycled through trials of normoxia and hypercapnia (21% $O_2$; 5% $CO_2$; 74% $N_2$) and compared the mean number of seizures observed during the two conditions. Relative to normoxia, the number of spike-wave seizures was lower during 5% $CO_2$ (p = 0.0028, n = 8; **Figure 4B1, C**, **Table 1**); hypercapnia also induced a powerful respiratory response (p = $3.78 \times 10^{-5}$, n = 8; **Figure 4B2,3 and 4D**, **Table 2**). Hypercapnia neither changed the duration (normoxia: 5.6 ± 0.4 s; hypercapnia: 6.2 ± 0.4 s; p = 0.22, n = 8, **Table 3**) nor the frequency (normoxia: 6.1 ± 1.0 Hz; hypercapnia: 6.2 ± 0.4 Hz; p = 0.28, n = 8, **Table 4**) of individual spike-wave seizures. Blood gas measurements revealed that 5% hypercapnia increased $PaCO_2$ (p = 0.022, n = 6; **Figure 4E2**) and slightly acidified blood pH (p = 0.00063, n = 6; **Figure 4E3**, **Table 6**). These results provide further support for the hypothesis that the neural circuits that produce spike-wave seizures are $CO_2$ sensitive, and thus pH sensitive. Moreover, the results demonstrate that neither the mechanics of elevated ventilation nor increased arousal, is sufficient to provoke spike-wave seizures.

## Optogenetic stimulation of the retrotrapezoid nucleus provokes spike-wave seizures

In addition to inducing hyperventilation and hypocapnia, hypoxia also lowers $PaO_2$ (see **Figure 3H1**), an effect that stimulates the carotid body, the principal peripheral chemoreceptor that initiates hyperventilation during hypoxic conditions (**Lindsey et al., 2018**; **López-Barneo et al., 2016**; **Semenza and Prabhakar, 2018**). Carotid body activity recruits neurons of the nucleus tractus solitarius that then excite neurons of the central respiratory pattern generator to drive a respiratory response (**Guyenet, 2014**; **López-Barneo et al., 2016**). To evaluate the capacity of hyperventilation to provoke seizures in the absence of hypoxia (and, therefore, in the absence of carotid body activation), we utilized

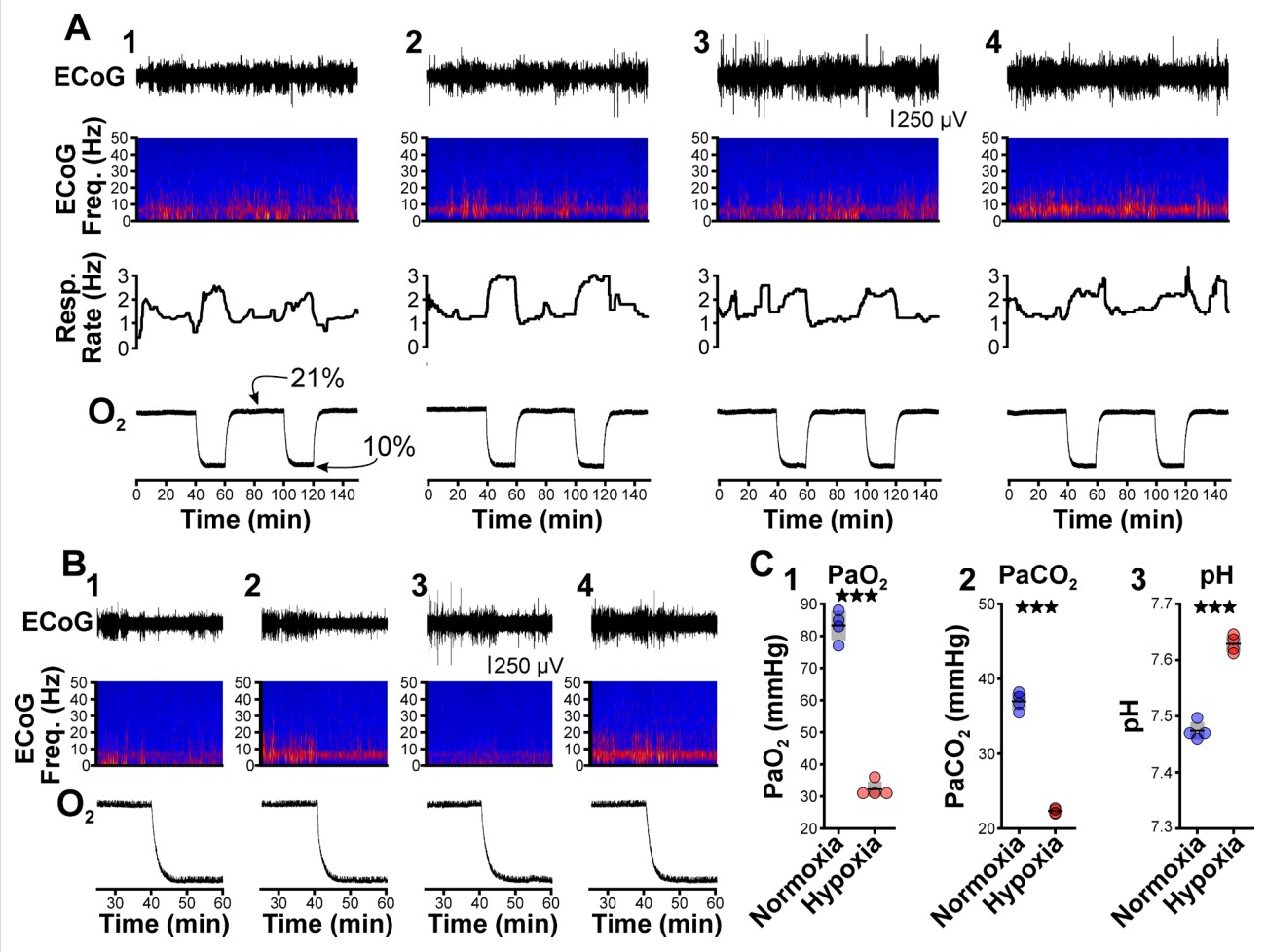

**Figure 2.** Hypoxia does not provoke hyperventilation-associated spike-wave seizures (SWS) in Wistar rats. (**A**) Plethysmography chambers recorded ventilation and electrocorticography/electromyography (ECoG/EMG) signals in four Wistar rats exposed to normoxia (i.e. 21% $O_2$) and hypoxia (i.e. 10% $O_2$). Panels 1–4 include responses from four Wistar rats, respectively, and show from top to bottom: ECoG, ECoG power spectrogram, respiratory rate, and chamber $O_2$. During the 2.5 hr recording session, rats were challenged twice with hypoxia. No SWS were observed during either normoxia or hypoxia. (**B**) Expanded views of the first transition from normoxia to hypoxia shown in panel A. Increased low frequency power during normoxia in some rats (e.g. panel B2) represents sleep. Hypoxia in Wistar rats generally increased arousal. (**C**) Arterial measurements in the same rats show that hypoxia challenges produced a predictable drop in arterial (1) $O_2$ and (2) $CO_2$, as well as (3) alkalosis. See *Table 5* for detailed statistics. ***p < 0.001.

an alternative approach to induce hyperventilation. Under physiological conditions, chemosensitive neurons of the retrotrapezoid nucleus (RTN), a brainstem respiratory center, are activated during an increase in $PaCO_2$ and a consequent drop in arterial pH (*Guyenet et al., 2016*; *Guyenet et al., 2019*; *Guyenet and Bayliss, 2015*) that then stimulate respiration. Optogenetic activation of RTN neurons in

**Table 5.** Arterial measurements in Wistar rats.

| Figure | Parameter | Comparison | Value | n | p Value |
|---|---|---|---|---|---|
| | | Normoxia | 83.25 ± 2.32 | | |
| 2C1 | $PaO_2$ | Hypoxia | 32.25 ± 1.25 | 4 | 0.0002 |
| | | Normoxia | 37.0 ± 0.59 | | |
| 2C2 | $PaCO_2$ | Hypoxia | 22.33 ± 0.16 | 4 | $6.6 \times 10^{-5}$ |
| | | Normoxia | 7.47 ± 0.01 | | |
| 2C3 | pH | Hypoxia | 7.63 ± 0.01 | 4 | $4.5 \times 10^{-5}$ |

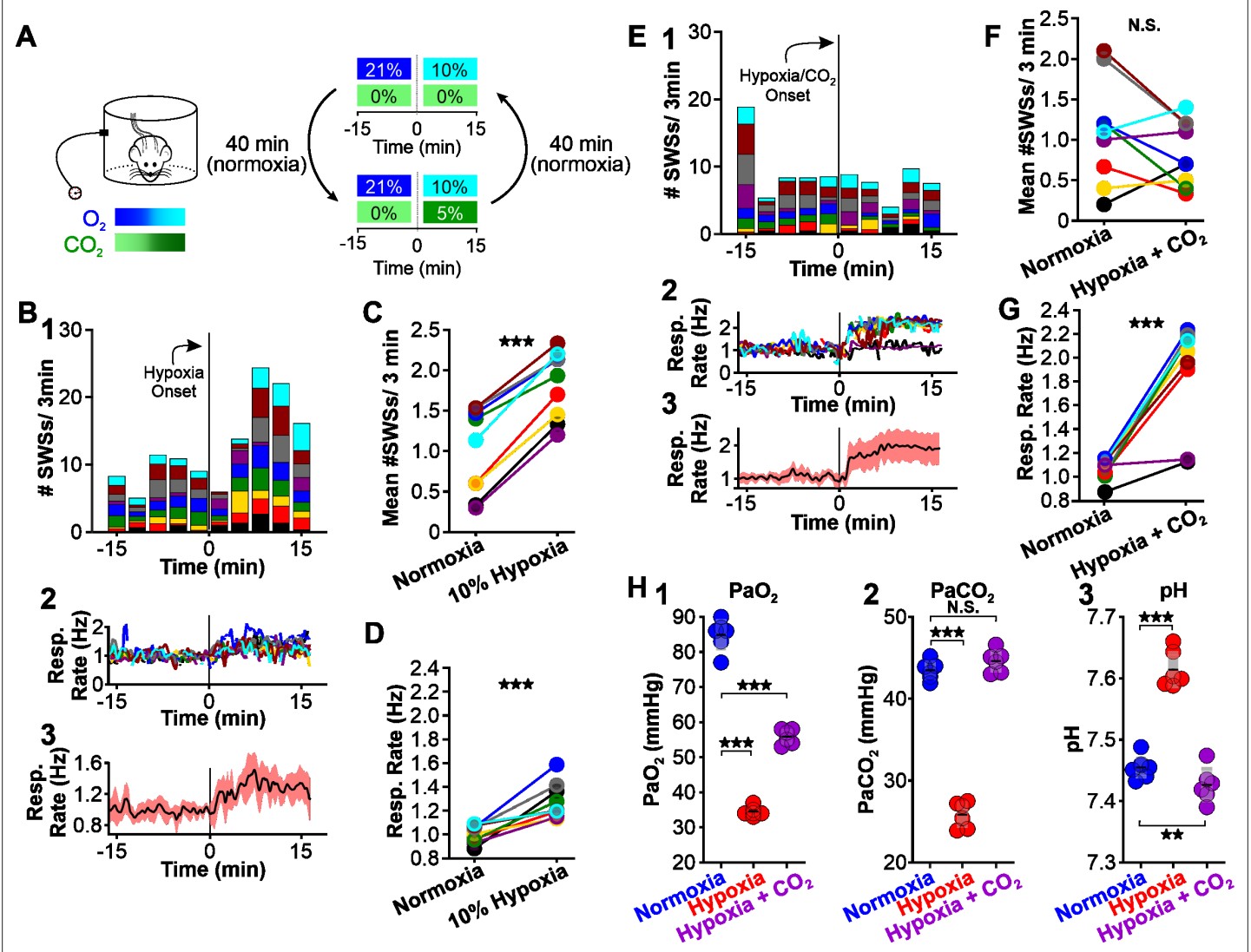

**Figure 3.** $CO_2$ suppresses hypoxia-provoked spike-wave seizures (SWS). (**A**) Experimental approach. Plethysmography chambers recorded ventilation and ECoG/EMG signals in WAG/Rij rats exposed to normoxia (i.e. 21% $O_2$) and then alternately challenged with hypoxia (i.e. 10% $O_2$) or hypoxia + $CO_2$ (i.e. 10% $O_2$, 5% $CO_2$). (**B–D**) Hypoxia challenge. (**B**) SWS and respiration quantification. (1) Stacked histogram illustrating SWS count for each animal before and after the onset of hypoxia. (2) Corresponding respiratory rate for each animal shown in panel B1. (3) Mean respiratory rate for all animals. (**C**) Mean SWS count per bin and (**D**) respiratory rate before and after hypoxia exchange. (**E–G**) Hypoxia + $CO_2$ challenge. (**E**) SWS and respiration quantification. (1) Stacked histogram illustrating SWS count for each animal before and after the onset of hypoxia + $CO_2$. (2) Corresponding respiratory rate for each animal shown in panel E1. (3) Mean respiratory rate for all animals. (**F**) MeanSWS count per bin and (**G**) respiratory rate before and after hypoxia + $CO_2$ exchange. (**H**) Arterial measurements show that hypoxia produced a predictable drop in arterial (1) $O_2$ and (2) $CO_2$, as well as (3) respiratory alkalosis (as in Wistar rats). Supplementing the chamber with 5% $CO_2$ normalizes arterial $CO_2$ and pH. Elevated arterial $O_2$ during hypoxia + $CO_2$ relative to hypoxia reflects a powerful inhalation response during the former condition (c.f. panels **D** and **G**). See **Tables 1, 2 and 6** for detailed statistics. **p < 0.01, ***p < 0.001.

normoxia is sufficient to evoke a powerful hyperventilatory response that alkalizes the blood (**Abbott et al., 2011**; **Souza et al., 2020**). Importantly, PaO₂ remains stable (or is slightly elevated) during opto-genetically induced respiration. Therefore, hyperventilation evoked by optogenetic RTN activation during normoxia both (1) promotes respiratory alkalosis without hypoxia and (2) is a more clinically relevant approximation of voluntary hyperventilation than hypoxia-induced hyperventilation.

We selectively transduced RTN neurons of WAG/Rij rats with a lentiviral approach using the PRSX8 promoter to drive channelrhodopsin expression (**Abbott et al., 2009**; **Hwang et al., 2001**; **Lonergan et al., 2005**; **Figure 5A and B**). Once channelrhodopsin was expressed, we challenged rats with two test trials: RTN photostimulation during normoxia and RTN photostimulation during hypercapnia

**Table 6.** Arterial measurements in WAG/Rij rats.

| Figure | Parameter | Comparison | Value | n | p Value |
|---|---|---|---|---|---|
| | | Normoxia | 84.93 ± 1.82 | | |
| | | Hypoxia | 34.50 ± 0.56 | 6 | $6.0 \times 10^{-6}$ |
| | | Normoxia | 84.93 ± 0.02 | | |
| 3H1 | $PaO_2$ | Hypoxia +$CO_2$ | 55.83 ± 0.87 | 6 | 0.000134 |
| | | Normoxia | 43.48 ± 0.47 | | |
| | | Hypoxia | 25.83 ± 0.65 | 6 | $2.1 \times 10^{-6}$ |
| | | Normoxia | 43.48 ± 0.47 | | |
| 3H2 | $PaCO_2$ | Hypoxia +$CO_2$ | 44.60 ± 0.55 | 6 | 0.42 |
| | | Normoxia | 7.45 ± 0.01 | | |
| | | Hypoxia | 7.61 ± 0.01 | 6 | $7.0 \times 10^{-6}$ |
| | | Normoxia | 7.45 ± 0.01 | | |
| 3H3 | pH | Hypoxia +$CO_2$ | 7.43 ± 0.01 | 6 | 0.008 |
| | | Normoxia | 84.93 ± 1.82 | | |
| 4E1 | $PaO_2$ | 5% $CO_2$ | 34.50 ± 0.56 | 6 | 0.00019 |
| | | Normoxia | 43.48 ± 0.47 | | |
| 4E2 | $PaCO_2$ | 5% $CO_2$ | 25.83 ± 0.65 | 6 | 0.022 |
| | | Normoxia | 7.45 ± 0.01 | | |
| 4E3 | pH | 5% $CO_2$ | 7.42 ± 0.01 | 6 | 0.00063 |

(*Figure 5C*); in a subset of animals, we cycled rats between the two conditions. In both trials, laser stimulation was delivered with trains of stimuli. During each train, the laser was pulsed at 20 Hz (10 ms pulse) for 2 s. The laser was then off for 2 s (i.e. intertrain interval = 2 s, see *Figure 5C*). This train stimulus was repeated for 15 min. Laser stimulation during normoxia provoked spike-wave seizures (p = 0.002, n = 10; *Figure 5D, E1, and F*, *Table 1*) and also increased ventilation (p = 0.019, n = 10; *Figure 5E2,3, and 5G*). Laser stimulation during normoxia did not alter the duration of individual spike-wave seizures (normoxia: 4.3 ± 0.6 s; normoxia-laser: 4.6 ± 0.5 s; p = 0.51, n = 10, *Table 3*). By contrast, the frequency of individual spike-wave seizures was lower during laser stimulation, relative to normoxia-alone (normoxia: 7.5 ± 0.2 Hz; normoxia-laser: 4.6 ± 0.5 Hz; p = $2.2 \times 10^{-4}$, n = 10, *Table 4*). Laser stimulation during hypercapnia in the same animals did not alter spike-wave seizure count (p = 0.86, n = 6; *Figure 5H1 and I*, *Table 1*), despite the induction of a strong hyperventilatory response (p = 0.031, n = 6; *Figure 5H2,3, and 5J*, *Table 2*). We observed no difference in duration (normoxia: 6.8 ± 1.0 s; hypercapnia-laser: 6.9 ± 1.0 s; p = 0.88, n = 6, *Table 3*) or frequency (normoxia: 7.9 ± 0.2 Hz; hypercapnia-laser: 6.9 ± 1.0 Hz; p = 0.33, n = 6, *Table 4*) of individual spike-wave seizures during normoxia versus hypercapnia coupled with laser stimulation. In sum, these results support the hypothesis that respiratory alkalosis is necessary to provoke seizures during hyperventilation and excludes carotid body activation as a contributing factor.

## Hypoxia-induced hyperventilation activates neurons of the intralaminar thalamus

Thus far, our results demonstrated that respiratory alkalosis (i.e. hyperventilation that promotes a net decrease in $PaCO_2$) provokes spike-wave seizures in the WAG/Rij rat. Next, we sought to identify brain structures activated during respiratory alkalosis that may contribute to spike-wave seizure provocation. We used the neuronal activity marker cFos to identify such structures in WAG/Rij rats. To isolate activation specifically associated with respiratory alkalosis, we first administered ethosuximide (200 mg/kg, i.p.) to suppress spike-wave seizures; respiration and ECoG/EMG signals confirmed ventilatory responses and spike-wave seizure suppression. Ethosuximide-injected rats were exposed to

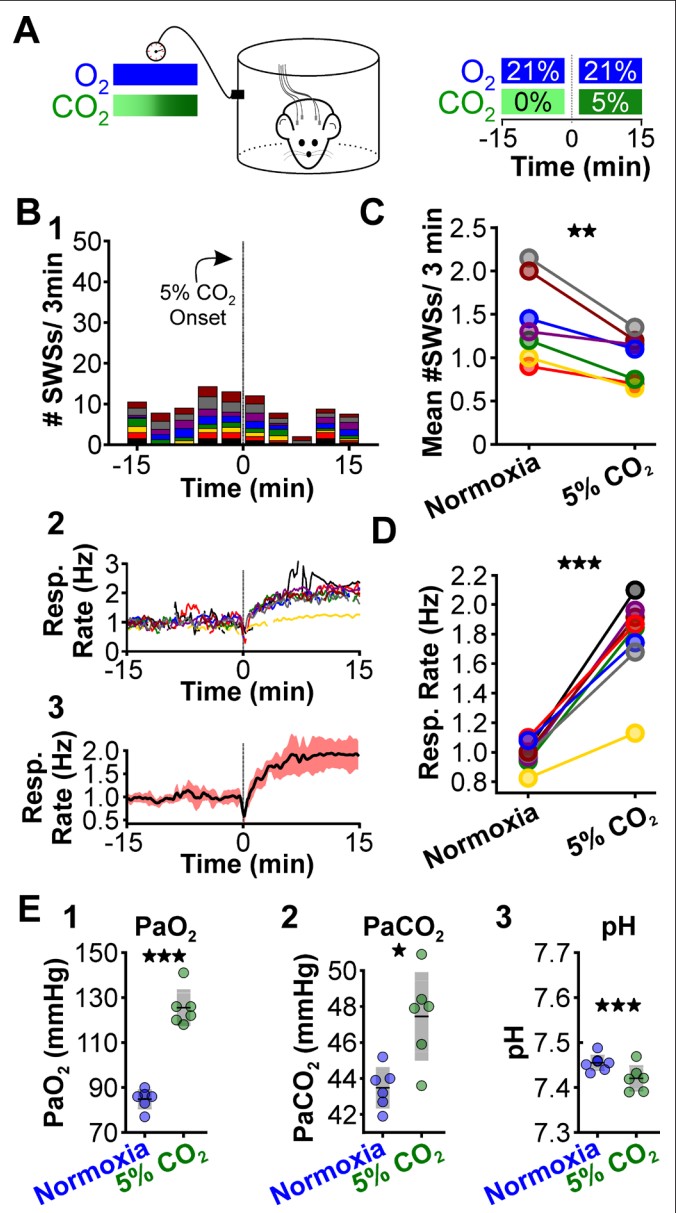

**Figure 4.** $CO_2$ suppresses spontaneous spike-wave seizures (SWS). (**A**) Experimental approach. Plethysmography chambers recorded ventilation and electrocorticography/electromyography signals in WAG/Rij rats exposed to normoxia (i.e. 21% $O_2$) and hypercapnia (i.e. 21% $O_2$, 5% $CO_2$). (**B**) SWS and respiratory quantification. (1) Stacked histogram illustrating SWS count for each animal before and after the onset of hypercapnia. (2) Corresponding respiratory rate for each animal shown in panel B1. (3) Mean respiratory rate for all animals. (**C**) Mean SWS count per bin and (**D**) respiratory rate before and after hypercapnia exchange. (**E**) Arterial measurements in the same rats show that hypercapnia produced a predictable increase in arterial (1) $O_2$ and (2) $CO_2$, as well as (3) respiratory acidosis. Increase arterial $O_2$ reflects robust ventilatory response during hypercapnia. See **Tables 1, 2 and 6** for detailed statistics. *p < 0.05, **p < 0.01, ***p < 0.001.

either hypoxia, normoxia, or hypoxia/hypercapnia for 30 min and then transcardially perfused 90 min later. Brains were harvested and evaluated for cFos immunoreactivity. Surprisingly, in rats exposed to hypoxia we observed heightened immunoreactivity in the intralaminar nuclei, a group of higher-order thalamic nuclei that, unlike first-order thalamic nuclei, do not receive peripheral sensory information (**Saalmann, 2014**; **Figure 6A and B**). Indeed, cFos immunoreactivity was largely absent from first-order thalamic nuclei and cortex, and was blunted in rats treated with normoxia and hypoxia/hypercapnia (**Figure 6B**). Importantly, the latter condition elevates respiration but normalizes arterial

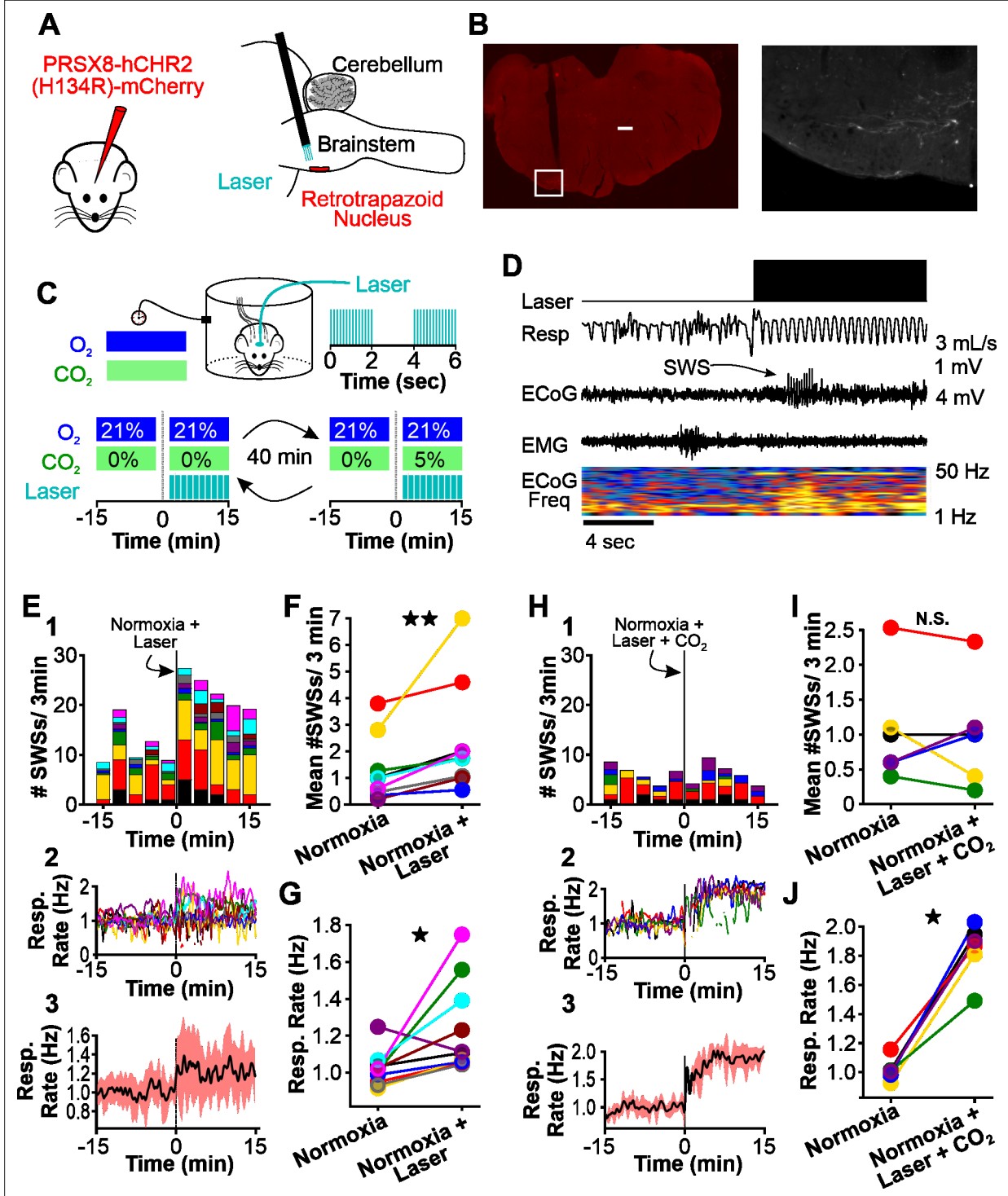

**Figure 5.** Normoxic hyperventilation provokes $CO_2$-sensitive spike-wave seizures (SWS). (**A**) Channelrhodopsin was virally delivered to the retrotrapezoid nucleus (RTN). The fiber optic cable was implanted during the surgery. After 3 weeks, photostimulation of the RTN induced hyperventilation. (**B**) After experimentation, opsin expression and fiber optic placement was verified. Representative image of mCherry-positive cells in the RTN. Large notch in slice is from optical fiber. Box on left image is enlarged on right image. Scale bar = 500 μm. (**C**) Experimental approach. Plethysmography chambers recorded ventilation and electrocorticography/electromyography signals in WAG/Rij rats exposed to normoxia (i.e. 21% $O_2$) and normoxia + $CO_2$ (i.e. 10% $O_2$, 5% $CO_2$). Channelrhodopsin-mediated photostimulation of the RTN was used to increase ventilation. (**D**) Example of ventilatory response and SWS during normoxic RTN photostimulation. (**E–G**) RTN photostimulation during normoxia. (**E**) SWS and respiration quantification. (1) Stacked histogram illustrating SWS count for each animal before and after normoxia photostimulation onset. (2) Corresponding respiratory rate for each

*Figure 5 continued on next page*

*Figure 5 continued*

animal shown in panel C1. (3) Mean respiratory rate for all animals. (**F**) Mean SWS count per bin and (**G**) respiration rate before and after normoxia photostimulation onset. (**H–J**) RTN photostimulation during hypercapnia (i.e. 21% $O_2$, 5% $CO_2$). (**H**) SWS and respiratory quantification. (1) Stacked histogram illustrating SWS count for each animal before and after hypercapnic photostimulation onset. (2) Corresponding respiratory rate for each animal shown in panel F1. (3) Mean respiratory rate for all animals. (**I**) Mean SWS count per bin and (**J**) respiratory rate before and after hypercapnic photostimulation onset. See *Tables 1, 2 and 6* for detailed statistics. *p < 0.05, **p < 0.01, not significant (n.s.).

pH (see *Figure 3G and H*). Immunoreactivity quantification revealed that the number of cFos-positive cells within the intralaminar thalamic nuclei was highest following hypoxia [ANOVA: $F_{(2, 6)}$ = 31.59, p = 0.00019, *Figure 6C*, *Table 7*].

As heightened cFos immunoreactivity was observed primarily following hypoxia that results in pronounced respiratory alkalosis, we next tested the hypothesis that neurons of the intralaminar nuclei are pH sensitive. We stereotaxically delivered the pan-neuronal expressing GCaMP7s (pGP-AAV-syn-jGCaMP7s-WPRE) to the intralaminar nuclei and harvested acute brain sections 3 weeks later (*Figure 6D*). Recording fluorescence changes in brain sections revealed that extracellular alkalosis quickly and reversibly activated neurons of the intralaminar nuclei (*Figure 6D*). An electrophysiological evaluation of pH sensitivity using voltage-clamp recordings ($V_{hold}$ = –50 mV) showed that alkaline bathing solutions evoke inward currents in intralaminar neurons (*Figure 6F and G*, *Table 7*), suggesting that excitatory ion channels and/or receptors were activated. Interestingly, alkaline-induced inward currents appeared blunted in other structures implicated in spike-wave seizure generation, such as somatosensory thalamus and cortex (*Figure 6H*). These results are consistent with previous reports of blunted, macroscopic pH sensitivity in the somatosensory thalamus (*Meuth et al., 2006*). Collectively, these results support the hypothesis that respiratory alkalosis activates pH-sensitive neurons of the intralaminar thalamic nuclei in the WAG/Rij rat.

## Discussion

Hyperventilation-provoked seizures associated with absence epilepsy were first formally described in 1928 by *Lennox, 1928* and despite the clinical ubiquity of utilizing hyperventilation to diagnose the common form of childhood epilepsy, no animal studies have attempted to resolve the physiological events that enable hyperventilation to reliably provoke spike-wave seizures. To resolve events and relevant brain structures recruited during this phenomenon, we first utilized the WAG/Rij rat to establish a rodent model that mimics hyperventilation-provoked spike-wave seizures in humans. With this model, we show that hyperventilation only provokes spike-wave seizures in seizure-prone, not generally seizure-free, rats. We then show that supplemental $CO_2$, by mitigating respiratory alkalosis, suppresses spike-wave seizures triggered by hyperventilation during either hypoxia or direct activation of brainstem respiratory centers. Moreover, supplemental $CO_2$, by producing respiratory acidosis, suppresses spontaneous spike-wave seizures (i.e. those occurring during normoxia) despite a compensatory increase in respiratory rate. These data demonstrate that spike-wave seizures are yoked to arterial $CO_2$/pH. Finally, we demonstrate that respiratory alkalosis activates neurons of the intralaminar thalamic nuclei, also in a $CO_2$-dependent manner; activation of these neurons is also pH sensitive. With these observations, we propose a working model wherein respiratory alkalosis activates pH-sensitive neurons of the intralaminar nuclei that in turn engage seizure-generating neural circuits to produce spike-wave seizures (*Figure 7*).

### Cortical EEG patterns evoked by hyperventilation

Hyperventilation produces stereotypical EEG patterns in both healthy children and children with absence epilepsy (*Barker et al., 2012*). In healthy children, hyperventilation can evoke an EEG pattern known as *hyperventilation-induced, high-amplitude rhythmic slowing* (HIHARS) that is often associated with altered awareness (*Barker et al., 2012*; *Lum et al., 2002*). Electrographically, HIHARS is distinct from spike-wave seizures insofar the EEG lacks epilepsy-associated spikes and resembles slow-wave sleep. Nonetheless, age-dependence and behavioral similarities between HIHARS and absence seizures exist (*Lum et al., 2002*; *Mattozzi et al., 2021*), thereby supporting the hypothesis that HIHARS and spike-wave seizures borrow from overlapping neural circuit mechanisms (*Mattozzi et al., 2021*). Indeed, while HIHARS and spike-wave seizures are clearly distinct EEG patterns, human

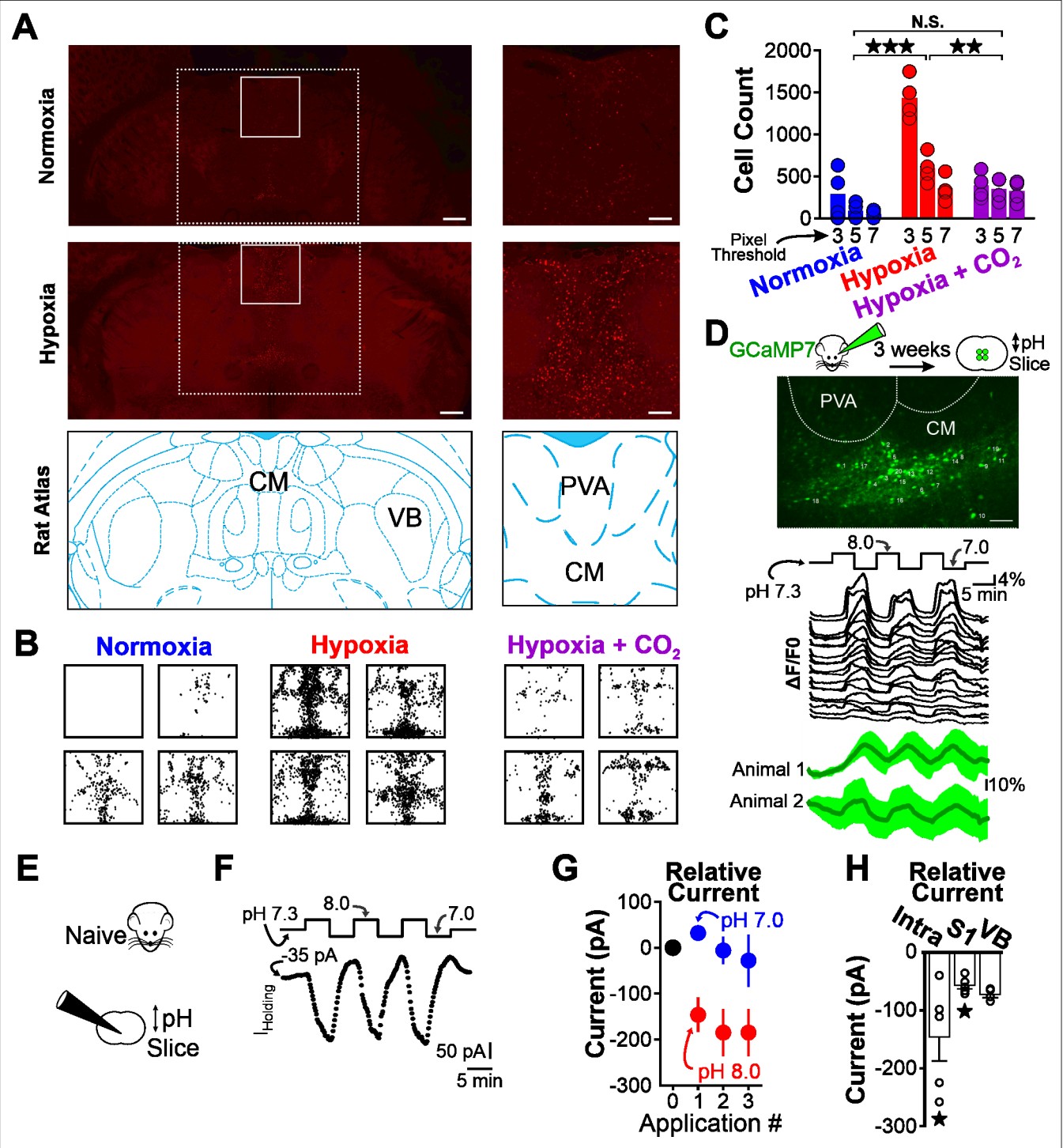

**Figure 6.** Hypoxia-induced hyperventilation activates intralaminar thalamic neurons. (**A**) cFos immunohistochemistry in horizontal sections of the WAG/ Rij rat. Dashed lines highlight the medial region of the thalamus containing the intralaminar nuclei. Solid lines demarcate regions containing elevated cFos expression and are expanded on right. Top images are collected from a rat exposed to 30 min of normoxia. Middle images are collected from a rat exposed to 30 min of hypoxia. Bottom images are taken from **Paxinos and Watson, 2007** and show the structural landmarks in the top and middle images. The central median nucleus (intralaminar thalamus) and ventrobasal complex (VB, first-order thalamus) are labeled. (**B**) cFos density plots show immunoreactivity in each of four rats exposed to either normoxia, hypoxia, or hypoxia + $CO_2$. Each black dot represents a cFos-positive cell, as identified with ImageJ (see Methods). Plots are aligned to expanded views in panel A. (**C**) Quantification of cFos labeled cells at different ImageJ thresholding values. (**D**) GCaMP7 was stereotaxically delivered to the intralaminar nuclei. Later, fluorescence changes were measured during extracellular alkaline challenges in acute slices containing the intralaminar nuclei. Individual ROIs show fluorescence changes during alkalosis (black

*Figure 6 continued on next page*

*Figure 6 continued*

traces). Mean responses from two animals are shown in green. The lag in response reflects the duration required for a complete solution exchange. (**E**) pH sensitivity of intralaminar neurons was also evaluated using electrophysiological measurements in acute brain slices. (**F**) Voltage-clamped intralaminar neurons ($V_{hold}$ = −50 mV) were exposed to control (pH 7.3), alkaline (pH 8.0), and acidic (pH 7.0) conditions. Inward currents were evoked during alkaline conditions. (**G**) Population intralaminar neuron response to alkaline conditions (n = 5). (**H**) Alkaline-evoked inward currents were largest in the intralaminar neurons (−146 ± 41.1 pA, n = 5), relative to similar measurements in neurons of the somatosensory cortex (S1, −59.1 ± 7.3 pA, n = 5) or thalamus (VB, ventrobasal nucleus, −68.1 ± 3.5 pA, n = 4). Inward currents during alkaline conditions (pH 8.0) in both intralaminar and S1 neurons were significantly larger, relative to their respective currents measured at a baseline pH of 7.3. Currents are presented as baseline-subtracted. **p < 0.01, ***p < 0.001. See *Table 7* for detailed statistics. Scale bars are 500 μm (*left*) and 100 μm (*right*).

spike-wave seizures observed during hyperventilation are subtly different from those occurring spontaneously (*Sadleir et al., 2008*), perhaps a reflection of the contribution of EEG-slowing circuitry to spike-wave seizures; while largely similar, we also found some differences in WAG/Rij spike-wave seizure frequency during some manipulations.

When viewed alongside work performed in the 1960s by *Sherwin, 1965*; *Sherwin, 1967*, our results support the hypothesis that hyperventilation-provoked spike-wave seizures and HIHARs share common circuits. Sherwin demonstrated that hyperventilation evokes HIHARS in cats (*Sherwin, 1965*), and that the stereotyped EEG pattern requires an intact central lateral nucleus of the thalamus (*Sherwin, 1967*). Together with the central medial (CM) and paracentral thalamic nuclei, the central lateral nucleus belongs to the anterior group of the intralaminar nuclei (*Saalmann, 2014*), the location

**Table 7.** cFos-positive cells in WAG/Rij rats.

| Figure | Threshold | Comparison | Counts (mean ± S.E.) | n | p Value |
|---|---|---|---|---|---|
| | | Normoxia | 282 ± 148.2 | | |
| | | Hypoxia | 1370 ± 137 | 4 | $1.5 \times 10^{-7}$ |
| | | Normoxia | 282 ± 148.2 | | |
| | | Hypoxia+ $CO_2$ | 385.5 ± 78.7 | 4 | 0.55 |
| | 3 | Hypoxia Hypoxia+ $CO_2$ | 1370 ± 137 385.5 ± 78.7 | 4 | $4.3 \times 10^{-7}$ |
| | | Normoxia | 112.3 ± 57.1 | | |
| | | Hypoxia | 595.3 ± 85.0 | 4 | 0.0005 |
| | | Normoxia | 112.3 ± 57.1 | | |
| | | Hypoxia+ $CO_2$ | 348 ± 68.9 | 4 | 0.045 |
| | 5 | Hypoxia Hypoxia+ $CO_2$ | 595.3 ± 85.0 348 ± 68.9 | 4 | 0.061 |
| | | Normoxia | 57.3 ± 29.2 | | |
| | | Hypoxia | 349 ± 75.0 | 4 | 0.021 |
| | | Normoxia | 57.3 ± 29.2 | | |
| | | Hypoxia+ $CO_2$ | 319.5 ± 63.1 | 4 | 0.036 |
| | 7 | Hypoxia Hypoxia+ $CO_2$ | 349 ± 75.0 319.5 ± 63.1 | 4 | 0.95 |
| 6C | | | | | |

| | | | Holding currents (pA) | N | P value |
|---|---|---|---|---|---|
| | Intra | Baseline pH 7.3 pH 8.0 | 9.9 ± 11.1 −136.6 ± 17.5 | 5 | 0.016 |
| | S1 | Baseline pH 7.3 pH 8.0 | 4.2 ± 5.3 63.3 ± 8.4 | 5 | 0.008 |
| 6H | VB | Baseline pH 7.3 pH 8.0 | 6.5 ± 19.4 −61.6 ± 27.4 | 4 | 0.057 |

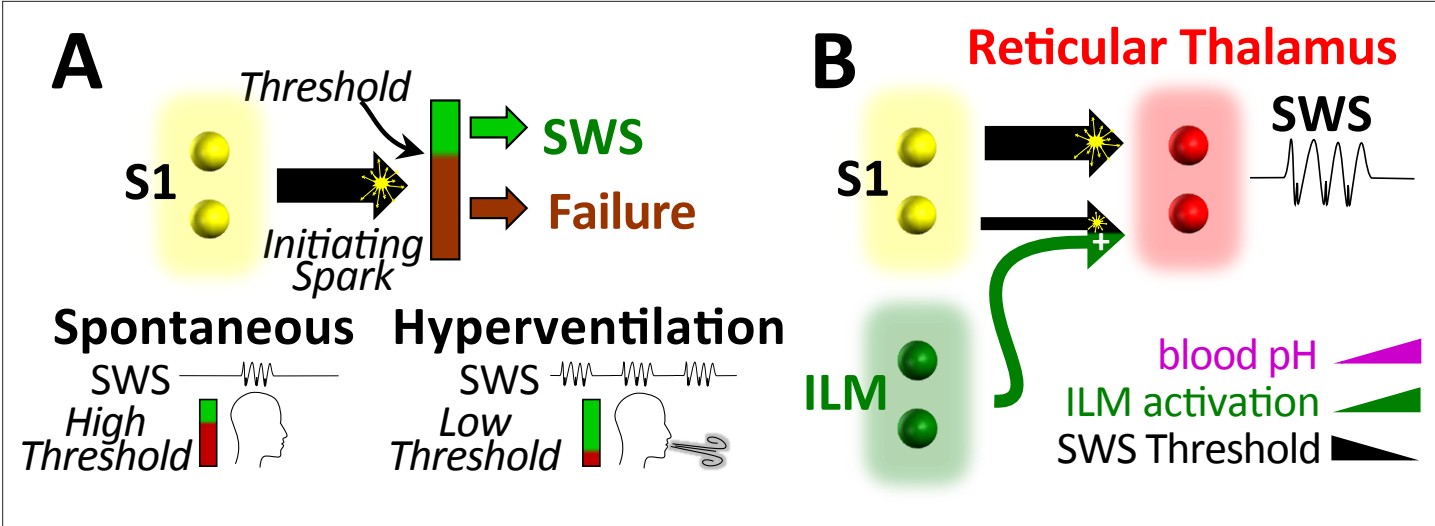

**Figure 7.** Working model. (**A**) Spike-wave seizures (SWS) only occur if initiating activity from S1 somatosensory cortex successfully overcomes a threshold, consistent with the cortical focus theory (*Meeren et al., 2002*). Hyperventilation-associated alkalosis reduces SWS threshold. (**B**) S1 initiating activity is proposed to overcome a seizure node formed by circuits in reticular thalamus to generate an SWS (*Paz and Huguenard, 2015*). We propose that hyperventilation-evoked respiratory alkalosis activates the intralaminar nuclei to reduce the threshold for S1 activity required to evoke an SWS. Thalamic pH sensitivity.

of cFos immunoreactivity associated with respiratory alkalosis and pH sensitivity (*Figure 6*). Indeed, at the time Sherwin postulated that the intralaminar nuclei of the thalamus are both chemoreceptive and capable of engaging widespread cortical activity (*Sherwin, 1967*). We now postulate that these nuclei are also instrumental for provoking spike-wave seizures during hyperventilation.

## Thalamocortical circuit involvement in spike-wave seizures

Decades of work have culminated in a canonical model wherein interconnected circuits between the cortex and thalamus support the initiation and maintenance of generalized spike-wave seizures (*Avoli, 2012*; *Beenhakker and Huguenard, 2009*; *Huguenard and McCormick, 2007*; *McCafferty et al., 2018*; *McCormick and Contreras, 2001*; *Meeren et al., 2002*). By recording from multiple sites in the WAG/Rij rat, Meeren et al. (*Meeren et al., 2002*) concluded that the peri-oral region of somato-sensory cortex provides the bout of hypersynchronous activity that initiates a spike-wave seizure. This activity then rapidly recruits additional somatosensory cortices and the lateral dorsal thalamus, a higher-order thalamic nucleus involved in spatial learning and memory (*Bezdudnaya and Keller, 2008*). Finally, first-order thalamic nuclei that encode somatosensory information (i.e. the ventrobasal complex) are recruited. This stereotyped succession of events occurs within the first 500 ms of the spike-wave seizure, after which the temporal relationships among cortical and thalamic structures are more unpredictable (*Meeren et al., 2002*). Additional studies support the hypothesis that cortical hyperactivity initiates spike-wave seizures (*Pinault, 2003*; *Pinault et al., 1998*) and have motivated what is generally referred to as the *cortical focus theory* for spike-wave seizure initiation (*Meeren et al., 2005*).

While resolving how seizures initiate and propagate through brain structures is of critical importance, such an understanding does not necessarily address the mechanisms that drive the highly rhythmic and hypersynchronous activity associated with ongoing spike-wave seizures. Extensive work on acute brain slice preparations clearly demonstrates that circuits between first-order thalamic nuclei and the reticular thalamic nucleus are sufficient to sustain rhythmic network activities, including those comparable to absence seizures (*Bal et al., 1995*; *Bal and McCormick, 1993*; *McCormick and Contreras, 2001*; *Krosigk von et al., 1993*). In this model, feedforward inhibition provided by retic-ular neurons evokes robust, hypersynchronous post-inhibitory rebound bursts among thalamocortical neurons that then relay activity back to reticular thalamus and to cortex. Reticular neuron-mediated feedforward inhibition of thalamocortical neurons, coupled with reciprocal excitation from thalam-ocortical neurons to reticular neurons, forms the basis of a rhythmogenic circuit that is proposed to

maintain spike-wave seizures. While this model very likely accounts for rhythmicity in the acute brain slice preparation, it is becoming less clear how first-order thalamocortical neurons actively contribute to the maintenance of spike-wave seizures recorded in vivo (*Huguenard, 2019*; *McCafferty et al., 2018*). Moreover, most current models of spike-wave initiation and maintenance neglect the potential contribution of the intralaminar nuclei to seizure initiation and maintenance despite several observations to the contrary.

In an effort to resolve structures capable of evoking spike-wave seizures, Jasper and colleagues electrically stimulated several thalamic nuclei in cats while recording EEG. By doing so in both lightly anesthetized (*Jasper and Droogleever-Fortuyn, 1947*) and unanesthetized (*Hunter and Jasper, 1949*) animals, the authors concluded that stimulation of the anterior intralaminar nuclei (i.e. central lateral, central medial, and paracentral nuclei) was sufficient to evoke spike-wave seizures that outlasted the stimulus; stimulation also produced behavioral repertoires associated with absence seizures. However, stimulation of first-order thalamic nuclei did not evoke spike-wave seizures, nor did it evoke seizure-like behaviors. Consistent with these observations, lesions to the intralaminar nuclei abolish pharmacologically induced spike-wave seizures in Sprague-Dawley rats (*Banerjee and Snead, 1994*); seizures persist following lesions to first-order nuclei. More recently, an EEG-fMRI study in human patients also implicates the intralaminar nuclei in the initiation of spontaneous spike-wave seizures (*Tyvaert et al., 2009*). Regrettably, (*Meeren et al., 2002*) did not include intralaminar thalamic recordings during their study of spike-wave seizure propagation in the WAG/Rij rat. Nonetheless, proposing the hypothesis that the intralaminar nuclei, not cortical structures, initiate spike-wave seizures, including those occurring spontaneously (i.e. not during hyperventilation), seems premature. Indeed, the possibility that activation of cortically projecting intralaminar neurons during hyperventilation recruits cortical structures to, in turn, initiate spike-wave seizures is equally plausible. In this model, respiratory alkalosis activates intralaminar neurons that, in turn, directly recruit spike-wave seizure initiation sites in the cortex. Alternatively, activated intralaminar neurons may increase the excitability of the reticular thalamic nucleus, a highly interconnected thalamic hub (*Swanson et al., 2019*), thereby lowering the threshold required for cortical input to spark a spike-wave seizure (see *Figure 7*). In support of this latter model, (*Purpura and Cohen, 1962*) demonstrated that electrical stimulation of the intralaminar nuclei evokes robust excitatory and inhibitory responses in the ventral thalamic nuclei.

First-order thalamic neurons express several pH-sensitive ion channels and receptors. TASK-1 and TASK-3, two TWIK-related acid-sensitive potassium channels, with the hyperpolarization-activated cyclic nucleotide–gated (HCN) ion channel, collectively play a critical role in stabilizing the resting membrane potential of first-order thalamic neurons (*Meuth et al., 2003*; *Meuth et al., 2006*). When activated, TASK channels hyperpolarize the membrane potential of thalamocortical neurons. In contrast, HCN channels depolarize thalamocortical neuron membrane potential. As extracellular acidification inhibits the activity of both channels, the opposing actions of TASK and HCN channels are simultaneously downregulated to yield no net effect on thalamocortical neuron membrane potential (*Meuth et al., 2006*), thereby stabilizing the membrane potential during acidic conditions. While not yet directly tested, the opposing actions of TASK and HCN channels also presumably stabilize thalamocortical membrane potential during alkaline conditions. Thus, while first-order thalamocortical neurons express pH-sensitive ion channels, these neurons are presumed to maintain stable membrane potentials during extracellular pH fluctuations. If true, then first-order thalamic nuclei are unlikely to support an active role in initiating hyperventilation-provoked spike-wave seizures. The extent to which higher-order thalamic nuclei express TASK and HCH channels remains unknown.

Importantly, intralaminar neurons recruited during hyperventilation-mediated alkalosis may not reflect intrinsic pH sensitivity. Instead, activation of intralaminar neurons during alkalosis may result from increased excitatory synaptic input. Intralaminar neurons receive significant, monosynaptic excitation from the midbrain reticular formation (*Ropert and Steriade, 1981*; *Steriade and Glenn, 1982*); first-order thalamic nuclei only do so negligibly (*Edwards and de Olmos, 1976*). Several reticular nuclei are critically important for respiration (*Guyenet and Bayliss, 2015*; *Smith et al., 2013*) and therefore provide clear rationale for testing the hypothesis that reticular-mediated excitation of the intralaminar nuclei drives hyperventilation-associated cFos expression (*Figure 6*). Notably, cFos expression was only observed during respiratory alkalosis (i.e. hypoxia) and not during hyperventilation associated with a normalized arterial pH (i.e. hypoxia-hypercapnia; c.f. *Figures 3H and 6B*). Thus, if reticular-mediated excitation of intralaminar neurons plays a role in hyperventilation-provoked spike-wave

seizures, then it does so only during conditions of respiratory alkalosis. Finally, the possibility that the synaptic terminals of intralaminar-projecting afferents are pH sensitive also warrants examination. Notably, solute carrier family transporters shuttle $H^+$ and $HCO_3^+$ across neuronal membranes and are proposed to regulate seizures, including spike-wave seizures (*Cox et al., 1997*; *Sander et al., 2002*; *Sinning and Hübner, 2013*). Alkaline conditions enhance excitatory synaptic transmission, an effect attributed to Slc4a8, an $Na^+$-driven $Cl^-$/bicarbonate exchanger (*Sinning et al., 2011*; *Sinning and Hübner, 2013*), that is expressed in the presynaptic terminals of excitatory neurons, including those in the thalamus (*Lein et al., 2007*). Thus, the enhancement of synaptic excitation onto intralaminar neurons remains a plausible mechanism to explain the large excitatory currents activated by alkalinization, as observed in *Figure 6*. The intralaminar nuclei appear particularly well suited to transduce alkalization into spike-wave seizures as pH sensitivity within these structures appears heightened relative to other nodes within the spike-wave seizure-generating circuitry (see *Figure 6H*).

## Conclusion

In aggregate, our data support the hypothesis that spike-wave seizures are yoked to arterial pH. The observation that respiratory alkalosis activates intralaminar thalamic neurons, and that such neurons are activated by alkaline conditions, reignites a 70-year-old hypothesis wherein intralaminar neurons actively participate in the initiation and maintenance of spike-wave seizures.

# Materials and methods
## Study Design

The goal of this study was to parameterize the effect of blood gases on spike-wave seizures. To do so, we adapted a clinically observed human phenomenon in absence epilepsy patients to a rodent model of spike-wave seizures. We demonstrate that spike-wave seizure occurrence correlates with rising or falling values of $PaCO_2$ and pH. Significantly, we show that neurons of the midline thalamus become activated after brief exposure to low $PaCO_2$ conditions. We propose that activity among pH-sensitive neurons in the thalamus, responsive to hyperventilation-induced hypocapnia, triggers spike-wave seizures. All physiology and ECoG/EMG recordings were performed in freely behaving WAG/Rij or Wistar rats. To reduce the number of animals, rats were exposed to multiple conditions. Experimenters were blinded to the condition for all respiration and ECoG/EMG data analysis. Group and sample size were indicated in the results section.

## Animals

All procedures conformed to the National Institutes of Health *Guide for Care and Use of Laboratory Animals* and were approved by the University of Virginia Animal Care and Use Committee (Charlottesville, VA, USA). Unless otherwise stated, animals were housed at 23–25°C under an artificial 12 hr light-dark cycle with food and water ad libitum. A colony of Wistar Albino Glaxo/from Rijswik (WAG/Rij rats) were kindly provided by Dr. Edward Bertram, University of Virginia and maintained in the animal facilities at The University of Virginia Medical Center. Male Wistar IGS Rats were purchased from Charles River (Strain Code: #003). Plethysmography, EEG, blood gas measurements, and c-Fos immunohistochemistry experiments were performed in 100+-day old WAG/Rij and Wistar rats as these ages correspond to when spike-wave seizures become robust in the WAG/Rij rat. Male and female rats were used in all experiments – no noticeable differences were observed. Of note, only male rats were used in optogenetic manipulations, as female rats were less likely to recover from surgery.

## Animal preparation

All surgical procedures were conducted under aseptic conditions. Body temperature was maintained at 37°C. Animals were anesthetized with 1–3% isoflurane or a mixture of ketamine (75 mg/kg), xylazine (5 mg/kg), and acepromazine (1 mg/kg) administered intra-muscularly. Depth of anesthesia was monitored by lack of reflex response to a firm toe and tail pinch. Additional anesthetic was administered during surgery (25% of original dose) if warranted. All surgeries, except the arterial catheter implantation, were performed on a stereotaxic frame (David Kopf Instruments, Tujunga, CA, USA). Postoperative antibiotic (ampicillin, 125 mg/kg) and analgesia (ketoprofen, 3–5 mg/kg, subcutaneously) were administered and as needed for 3 days. Animals recovered for 1–4 weeks before experimentation.

## Electrocorticogram (ECoG) and electromyography (EMG) electrode implantation

Commercially available rat recording devices were purchased from Plastics One (Roanoke, VA, USA). Recording electrodes were fabricated by soldering insulated stainless-steel wire (A-M system, Sequim, WA, USA) to stainless-steel screws (Plastics One) and gold pins (Plastics One). On the day of surgery, a small longitudinal incision was made along the scalp. Small burr holes were drilled in the skill and ECoG recording electrodes were implanted bilaterally in the cortex. Reference electrodes were placed in the cerebellum. A twisted looped stainless-steel wire was sutured to the superficial neck muscles for EMG recordings. The recording device was secured to the skull with dental cement and incisions were closed with absorbable sutures and/or steel clips.

## PRSX-8 lentivirus preparation

The lentivirus, *PRSX8*-hCHR2(H134R)-mCherry, was designed and prepared as described previously (*Abbott et al., 2009*). Lentivirus vectors were produced by the Salk Institute Viral Vector Core. The titer for the *PRSX8*-hCHR2(H134R)-mCherry lentivirus was diluted to a working concentration of $1.5 \times 10^{10}$ TU/mL. The same batch of virus was used for all experiments included in this study.

## Virus injection and fiber optic ferrule implantation

Borosilicate glass pipettes were pulled to an external tip diameter of 25 µm and backfilled with the lentivirus, *PRSX8*-hCHR2(H134R)-mCherry. Unilateral virus injections in the RTN were made under electrophysiological guidance of the antidromic potential of the facial nucleus (see *Abbott et al., 2009*; *Souza et al., 2018*). A total of 400 nL was delivered at three rostro caudal sites separated by 200 or 300 µm in the RTN. Illumination of the RTN was performed by placing a 200-µm-diameter fiber optic (Thor Labs, #BFL37-200; Newton, NJ, USA) and ferrule (Thor Labs, #CFX128-10) vertically through the cerebellum between 300 and 1000 µm dorsal to RTN ChR2-expressing neurons. These animals were also implanted with ECoG/EMG recording electrodes, as detailed above. All hardware was secured to the skill with dental cement. Animals recovered for 4 weeks, as this provided sufficient time for lentivirus expression in the RTN. Virus injection location was verified post-hoc. Only animals that responded to optical stimulation, demonstrated by an increase in respiratory frequency, were included in the results.

## Physiology experiments in freely behaving rats

All experiments were performed during the dark cycle (hours 0–4) at ambient room temperature of 27–28°C. Rats were habituated to experimental conditions for a minimum of 4 hr, 1–2 days before experiment start. On the day of recordings, rats were briefly anesthetized with 3% isoflurane for <5 min to connect the ECoG/EMG recording head stage to a recording cable and, when necessary, to connect the fiber optic ferrule to a fiber optic cord (multimode 200 µm core, 0.39 nA) attached to a 473 nm blue laser (CrystaLaser model BC-273–060 M, Reno, NV, USA). Laser power was set to 14 mW measured at the junction between the connecting fiber and the rat. Rats were then placed immediately into a whole-body plethysmography chamber (5 L, EMKA Technologies, Falls Church, VA, USA). Recordings began after 1 hr of habituation. The plethysmography chamber was continuously perfused with room air or protocols cycling through specific gas mixtures of $O_2$, $N_2$, and $CO_2$ (total flow: 1.5 L/min). Mass flow controllers, operated by a custom-written Python script, regulated gas exchange. Respiratory flow was recorded with a differential pressure transducer. The respiratory signal was filtered and amplified at 0.1–100 Hz, X 500 (EMKA Technologies). Respiratory signals were digitized at 200 Hz (CED Instruments, Power1401, Cambridge, England). ECoG and EMG signals were amplified (X1000, Harvard Apparatus, Holliston, MA, USA; Model 1700 Differential Amplifier, A-M Systems), bandpass filtered (ECoG: 0.1–100 Hz; EMG: 100–300 Hz), and digitized at 200 Hz. Respiratory flow, ECoG/EMG recordings, $O_2$ flow, and the laser pulse protocol were captured using Spike2, 7.03 software (CED Instruments).

Spike-wave seizures were manually identified by blinded individuals. Once identified, custom Matlab scripts identified the true onset and offset of each spike-wave seizure by locating the time point of the first and last peak of the seizure (as defined by sections of the recording that were 2.5 times the pre-seizure RMS baseline); seizure duration was defined as the duration between the first and last peak. Seizure frequency was quantified by computing a fast Fourier transform (FFT) on the

event. Spike-wave seizure occurrence before and during specific conditions is shown as a peri-stimulus time histogram aligned at time = 0 at gas exchange onset or laser-on for optogenetic stimulations. Spike-wave seizure counts were quantified in three bins beginning ±15 min of gas exchange or laser onset. Total spike-wave seizure counts were obtained by summing the number of spike-wave seizures between –15 and 0 min (control) and 0 and +15 min (manipulation). Respiratory frequency ($f_R$, in breaths/minute) was derived from the respiration trace. The respiration trace was divided into individual windows, each 10 s in duration, and an FFT was computed on each discrete window. The respiratory rate for each window was defined by the FFT frequency with the maximal power density. Once derived for each window, we then applied a 30 s moving average to smooth the trace. RTN neurons were optically stimulated with 10 ms pulses delivered at 20 Hz for 2 s, followed by 2 s rest. This stimulation protocol was repeated for 20 min.

## Femoral artery catheterization, blood gases and pH measurements

Arterial blood samples for blood gas measurements through an arterial catheter during physiological experiments. One day prior to the experiments, rats anesthetized with isoflurane (2% in pure $O_2$) and a polyethylene catheter (P-10 to P-50, Clay Adams, Parsippany, NJ, USA) was introduced into the femoral artery by a small skin incision toward the abdominal aorta. The catheter was then tunneled under the skin and exteriorized between the scapulae with two inches of exposed tubing anchored with a suture. On the day of the experiment, animals were briefly anesthetized with 1–2% isoflurane to attach tubing for blood collection before placement into the plethysmography recording chamber. Arterial blood gases and pH were measured using a hand-held iStat configured with CG8+ cartridges (Abbott Instruments, Lake Bluff, USA).

## cFos histology

After exposing WAG/Rij rats to 30 min of hypoxia (10% $O_2$; 90% $N_2$) or hypoxia/hypercapnia (10% $O_2$; 5% $CO_2$; 75% $N_2$) rats were deeply anesthetized and perfused transcardially with 4% paraformaldehyde (pH 7.4). Brains were removed and post-fixed for 12–16 hr at 4°C. 40 μm horizontal sections of the thalamus (D/V depth –5.3 to 6.0 mm) were obtained using a Leica VT 1000 S microtome (Leica Biostystems, Buffalo Grove, IL, USA) and collected in 0.1 M phosphate buffer (PB) with 0.1% sodium azide (Millipore-Sigma, St. Louis, MO, USA). Sections were then transferred to a 0.1 M PB solution containing 20% sucrose for 1 hr, snap-frozen and transferred to 0.1% sodium borohydride for 15 min. Slices were washed 2× in phosphate buffered saline (PBS). All blocking and antibody solutions were prepared in an incubation buffer of 0.1% sodium azide, 0.5% Triton X-100%, and 2% normal goat serum. Sections were blocked for 4 hr at room temperature or overnight at 4°C in incubation buffer. Sections were washed 3× with PBS between primary and secondary antibody solutions. Primary antibody solutions containing rabbit anti-cFos (1:2000; Cell Signaling Technology Cat# 2250, RRID: AB_2247211, Danvers, MA, USA) and biotin (1:200, Jackson ImmunoResearch, West Grove, PA; RRID: AB_2340595) were prepared in incubation buffer and incubated overnight at 4°C. Sections were then incubated overnight in secondary antibody solutions containing donkey strepavidin-Cy3 (1:1000, Jackson ImmunoResearch; RRID: AB_2337244). Immunohistochemical controls were run in parallel on spare sections by omitting the primary antisera and/or the secondary antisera. Sections from each well were mounted and air-dried overnight. Slides were cover-slipped with VectaShield (VectorLabs, Burlingame, CA) with the addition of a DAPI counterstain. All images were captured with a Z1 Axioimager (Zeiss Microscopy, Thornwood, NY, USA) with computer-driven stage (Neurolucida, software version 10; MicroBrightfield, Inc, Colchester, VT, USA). Immunological sections were examined with a 10× objective under epifluorescence (Cy3). All sections were captured with similar exposure settings. Images were stored in TIFF format and imported into ImageJ (NIH). Images were adjusted for brightness and contrast to reflect the true rendering as much as possible. To count cFos-positive cells, we utilized the particle analysis tools in ImageJ, and applied a pixel area threshold of varying stringency (0–7px$^2$). Repeated measures ANOVAs for each treatment and threshold were used for statistical analyses.

## Calcium imaging

pGP-AAV-syn-jGCaMP7s-WPRE (Addgene #104487-AAV9) was stereotaxically delivered to the central median thalamic nucleus in P20-30 rats with sterile microliter calibrated glass pipettes. A picospritzer

(Picospritzer III, Parker Hannifin) was used to deliver 100–200 nl of virus. Three weeks later, animals were sacrificed and their brains harvested for acute brain slice preparation. Animals were deeply anesthetized with pentobarbital and then transcardially perfused with an ice-cold protective recovery solution containing the following (in mm): 92 NMDG, 26 NaHCO$_3$, 25 glucose, 20 HEPES, 10 MgSO$_4$, 5 Na-ascorbate, 3 Na-pyruvate, 2.5 KCl, 2 thiourea, 1.25 NaH$_2$PO$_4$, 0.5 CaCl$_2$, titrated to a pH of 7.3–7.4 with HCl (*Ting et al., 2014*). Horizontal slices (250 μm) containing the intralaminar thalamic nuclei were cut in ice-cold protective recovery solution using a vibratome (VT1200, Leica Biosystems) and then transferred to protective recovery solution maintained at 32–34°C for 12 min. Brain slices were kept in room temperature artificial cerebrospinal fluid (ACSF) containing (in mm): 3 KCl, 140 NaCl, 10 HEPES, 10 Glucose, 2 MgCl$_2$, 2 CaCl$_2$. The solution was bubbled with 100% O$_2$ and the pH was set by adding varied amounts of KOH. Fluorescence signals were measured with a spinning disk confocal microscope outfitted with an sCMOS camera (ORCA-Flash4.0, Hamamatsu, Bridgewater, NJ, USA).

## Voltage-clamp recordings

Brain slices were prepared as described above for calcium imaging experiments; similar ACSF solutions were also used. Thalamic neurons were visualized using a Zeiss Axio Examiner.A1 microscope (Zeiss Microscopy, Thornwood, NY, USA) and an sCMOS camera (ORCA-Flash4.0, Hamamatsu). Recording pipettes were pulled on a P1000 puller (Sutter Instruments) from thin-walled borosilicate capillary glass (Sutter Instruments, Novato, CA, USA). Pipettes (2–3 MΩ tip resistance) were filled with (in mM) 100 K-gluconate, 9 MgCl$_2$, 13 KCl, 0.07 CaCl$_2$, 10 HEPES, 10 EGTA, 2 Na$_2$ATP, 0.5 NaGTP, pH adjusted to 7.3 with KOH, and osmolality adjusted to 275 mOsm. Recordings were performed in the whole cell patch clamp configuration. Data were acquired in pClamp software (Molecular Devices, San Jose, CA, USA) using a Multiclamp 700B amplifier (Molecular Devices), low-pass filtered at 2 kHz, and digitized at 10 kHz (Digidata 1,440 A, Molecular Devices). Access resistance was monitored by repeatedly applying a –5 mV hyperpolarizing voltage step and converting the resultant capacitive transient response into resistance (*Ulrich and Huguenard, 1997*). A good recording consisted of an access resistance less than 20 MΩ that changed by less than 20% over the course of the recording; recordings that did not meet these criteria were discarded.

## Data analysis and statistics

Statistical analyses were performed in GraphPad Prism v7 (San Diego, CA, USA). All data were tested for normality before additional statistical testing. Statistical details, including sample size, are found in the results section and corresponding supplemental tables. Either parametric or non-parametric statistical analyses were performed. A significance level was set at 0.05. Data are expressed as mean ± SEM. Data have been deposited at https://doi.org/10.5061/dryad.zcrjdfncm and custom scripts are available at https://github.com/blabuva/eLife-2022-11-e72898 (*Beenhakker, 2022*; copy archived at swh:1:rev:182cf0b04ecc861aee0dacd271504fa8be7c7516).

# Additional information

### Funding

| Funder | Grant reference number | Author |
| --- | --- | --- |
| National Institute of Neurological Disorders and Stroke | R01NS099586 | Mark P Beenhakker |
| National Institute of Neurological Disorders and Stroke | R56NS099586 | Mark P Beenhakker |

The funders had no role in study design, data collection and interpretation, or the decision to submit the work for publication.

### Author contributions

Kathryn A Salvati, Conceptualization, Data curation, Formal analysis, Investigation, Methodology, Writing – original draft; George MPR Souza, Data curation, Methodology, Writing – review and

editing; Adam C Lu, Data curation, Investigation, Writing – review and editing; Matthew L Ritger, Data curation; Patrice Guyenet, Conceptualization, Writing – review and editing; Stephen B Abbott, Investigation, Methodology, Writing – review and editing; Mark P Beenhakker, Conceptualization, Funding acquisition, Investigation, Project administration, Resources, Supervision, Writing – review and editing

**Author ORCIDs**
Kathryn A Salvati (D) http://orcid.org/0000-0002-9557-9259
Stephen B Abbott (D) http://orcid.org/0000-0003-1244-3637
Mark P Beenhakker (D) http://orcid.org/0000-0002-4541-0201

**Ethics**
All procedures conformed to the National Institutes of Health Guide for Care and Use of Laboratory Animals and were approved by the University of Virginia Animal Care and Use Committee (protocol #3892).

**Decision letter and Author response**
Decision letter https://doi.org/10.7554/eLife.72898.sa1
Author response https://doi.org/10.7554/eLife.72898.sa2

---

## Additional files

**Supplementary files**
• Transparent reporting form

**Data availability**
All data generated or analysed during this study are included in the manuscript and corresponding data tables. We have also deposited our raw datasets for each figure with Dryad at the following URL: https://doi.org/10.5061/dryad.zcrjdfncm.

The following dataset was generated:

| Author(s) | Year | Dataset title | Dataset URL | Database and Identifier |
|---|---|---|---|---|
| Beenhakker MP | 2022 | Data from: Respiratory alkalosis provokes spike-wave discharges in seizure-prone rats | http://dx.doi.org/10.5061/dryad.zcrjdfncm | Dryad Digital Repository, 10.5061/dryad.zcrjdfncm |

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
