## [Editor Report]

The study evaluates the long debated question of how respiration affects seizure susceptibility. The authors use a rigorous approach to manipulate the gases breathed in by seizure prone rats while monitoring their respiration, electroencephalographic activity, blood pH and gas levels. They show that changes in pH caused by hyperventilation drive spike-wave seizures, that optogenetically driving hyperventilation induced spike-wave seizures by changing pH, and that intralaminar nuclei in the thalamus contain neurons that are activated during hyperventilation and are pH sensitive.

---

## [Decision Letter]

**Decision letter after peer review:**

Thank you for submitting your article "Respiratory Alkalosis Provokes Spike-Wave Discharges in Seizure- Prone Rats" for consideration by *eLife*. Your article has been reviewed by 2 peer reviewers, and the evaluation has been overseen by Joseph Gleeson as Reviewing Editor and Laura Colgin as the Senior Editor. The following individuals involved in review of your submission have agreed to reveal their identity: Chris G Dulla (Reviewer #1); William Nobis (Reviewer #2).

Essential revisions:

1) Readers will want to have a better appreciation for the Quantification of spike-wave seizure properties (duration, frequency components), providing post-hoc confirmation of viral targeting of the RTN.

2) The abstract should contain more information about the experiments utilizing channelrhodopsin to manipulate neural activity, and how this information was interpreted. As of now, the data could missed by many readers.

3) Readers will want some clarification to interpret the correlation between pH and electrical activity of intralaminar neurons.

4) More insight could be offered into the circuit mechanisms by which intralaminar neurons initiate or predispose circuits to generating spike-wave seizures.

5) Addressing the issue of whether intralaminar thalamus neurons are pH sensitive with further controls.

*Reviewer #1:*

– The conclusions of the optogenetic stimulation of RTN would be strengthened by showing post-hoc validation of the viral targeting in a spatially specific manner. Can the authors confirm that those experiments accurately targeted the RTN and not other regions more broadly?

– Do the manipulations (decreased O2, decreased 02/elevated C02, elevated C02 in normoxia, optogenetic stimulation of RTN) alter the duration or frequency components of SWSs? Or are the seizures identical across conditions, but their incidence is changed?

– The alignment of the pH manipulations and changes in GCaMP DF/F in Figure 6D are not clear. It appears that changes in DF/F are not time locked to changes in pH? Is that correct? If so what accounts for this time lag?

– Why is the change in respiration rate during optogenetic stimulation of the intralaminar thalamus so variable when stimulated during normoxia (Figure 5E) but so robust and uniform when done in high C02 (Figure 5H). Is this because the relationship between breathing rate the pH is different in different conditions or is it due to variability in targeting of the RTN?

– A better description of the cFos thresholding quantification should be included. It is not clear what was done there. A clearer description (visually in the figure and/or in the text) of the statistical comparisons made for each condition/threshold should also be included. This could be added as a supplemental figure if it make the presentation easier.

– The discussion would be strengthened by considering how increased input from intralaminar thalamic neurons trigger SWS. The authors nicely consider how intralaminar thalamic neuron activity may arise from increased excitatory input, but there is not much consideration of how increased output from intralaminar thalamic neurons feed into canonical SWS circuitry (nRT, VB, cortex).

*Reviewer #2:*

Salvati et al., sought to investigate the means by which hyperventilation can provoke generalized spike-wave discharges and seizures in a susceptible rat model. Hyperventilation is long established and used clinically to induce seizures in childhood absence epilepsy patients and has been thought to be related to the respiratory alkalosis induced by the event and in particular this pH enhancing thalamocortical synchronicity, but this had yet to be mechanistically worked out. The authors build on recent studies to perform an elegant set of experiments that implicate the respiratory alkalosis and arterial CO2 itself in the generation of seizures. They further establish that the intralaminar thalamic region may be in-part mediating this seizure activity.

The use of the optogenetically driven hyperventilation and respiratory alkalosis in normoxic conditions was a particularly strong experimental approach.

Overall, the authors conclusions are mostly supported by the data, but there remain some aspects of the methodology and data analysis that would better support the case.

There is a well-balanced discussion that does a good job of delineating the limits of the conclusions they are able to firmly establish by this work, however, it is somewhat disappointing that more of the mechanistic studies they suggest were not performed which limits the full impact of the work.

1) Regarding the optogenetic experiments, there are a few methodological clarifications and controls that need addressing, otherwise this is a powerful approach to isolate the effect of hyperventilation on spike waves and seizure induction.

2) The evaluation of the intralaminar thalamus activation by hypoxia-induced hyperventilation is exciting but the data does not yet support that these neurons are pH sensitive, rather they are activated in the context of changes in arterial CO2, and at times in the manuscript this is not well clarified. Alkaline pH can be a general neuronal activator, and further controls and comparisons to other thalamic regions would strengthen the results presented.

3) Finally, by not addressing experimentally key aspects of the hypothesis – that intralaminar thalamic activation driven by respiratory alkalosis drives SWS – the broader impact of this manuscript is limited.

Recommendations for the authors

Related to point 1 above:

– It would be helpful to see viral expression and cannula placement to determine if hyperventilation and SWS induction was truly due to activation of RTN neurons. The C1 region, amongst other medullary regions, are in close proximity to the RTN.

– There seems to be a disconnect in SWS generation and the level of hyperventilation reached – see 5D and 5E – some of the animals with the highest increase in SWS had a minimal change in respiratory rate. Do individual animals have a varying pH change related to hyperventilation (perhaps looking at respiratory rate change and pH and arterial CO2 in the animals from figure 3 would be illustrative) or is there another explanation that could be suggested?

– It's stated that only animals that responded to optical stimulation with an increase in respiratory frequency were included in the analysis shown in Figure 5. Did you determine a reason why some animals did not respond – such as poor viral targeting, expression or poor cannula targeting? it would be illuminative to provide the numbers for those that had accurate targeting but no response to optical stimulation. Also, it appears that one animal that had a decrease in respiratory frequency (the dark purple Figure 5E) was included in the analysis despite the exclusion criteria listed.

Regarding point 2 above:

– Did you quantify cfos activation in other thalamic regions of interest such as the reticular nucleus, was the intralaminar region the only region with significant activation?

– As for the calcium imaging, the sample imaging does not provide enough anatomical context for orientation. Also, as alkaline pH can be a general neuronal activator and we might expect results similar to what was obtained looking at a number of brain regions. To make the case that the intralaminar nucleus is particularly or selectively activated requires a control comparison. Perhaps the VB thalamus as it is relevant to the potential mechanisms and anatomically adjacent.

– Was cfos activation driven by RTN opto stimulation evaluated?

The discussion nicely discusses the current extant literature and the context of this study as well as the limitations of this manuscript.

It would enhance the impact of the study if some of these points were addressed experimentally, as currently while the manuscript very elegantly and nicely supports that respiratory alkalosis from hyperventilation drives seizure induction, the critical nature and the role of the intralaminar thalamus is less supported – is it truly necessary for the generation of spike wave discharges. Also whether it has a direct or more indirect – potentiated by other inputs – role in chemosensation is not clear.

– Patch-clamp electrophysiology of intralaminar neurons could readily determine if there is any intrinsic pH sensitivity of these neurons, looking for a change in firing due to changes in pH. Answering the question of whether this is presynaptically (maybe from reticular nuclei input) could also be addressed by looking at changes in spontaneous transmission.

– Following electrophysiologic studies up with evaluation of expression of pH sensitive ion channels/receptors would be important as noted in your discussion.

– It would be interesting to cfos expression in response to respiratory alkalosis in reticular regions and the dorsal raphe – which may be more chemosensitive and project to the intralaminar nucleus.

– While this may be beyond the scope of the manuscript, optogenetic activation of the intralaminar nucleus to produce spike wave discharges and seizures, and also doing this in the context of hypoxia-induced hyperventilation or rescue with excessive CO2 would be very powerful and more fully evaluate the model in Figure 7.

---

## [Author Response]

Essential revisions:1) Readers will want to have a better appreciation for the Quantification of spike-wave seizure properties (duration, frequency components), providing post-hoc confirmation of viral targeting of the RTN.

We agree that providing a more quantitative description of spike-wave seizure properties is warranted. We now include these details in our revised manuscript. Interestingly, the duration of individual spike-wave seizures observed before and during the various forms of induced hyperventilation are nearly always the same (except during hypoxia/hypercapnia). The frequencies of individual seizures before and during hyperventilation are also generally similar, although during hypoxia/optogenetic stimulation we did find that seizures are ~20% slower; hypercapnia did not alter seizure frequency. Currently, we can only speculate that any seizure slowing might reflect mechanisms that also contribute to hyperventilation-induced high amplitude, rhythmic slowing (HIHARS) observed in healthy patients.

Unfortunately, very few human studies have systematically compared features of spontaneous versus hyperventilation-induced spike-wave seizures. However, we did find one 2008 study published in Epilepsia (Sadleir et al., 2008, Epilepsia 49(12):2100-7). Therein, the authors evaluate spike-wave seizures that occur during various behavioral states, as well as during hyperventilation, and conclude that subtle differences exist. For example, in humans, hyperventilation-induced spike-wave seizures are slightly longer in duration than those occurring spontaneously. However, differences in spike-wave seizures observed across the patient population are generally larger than those observed during different behavioral states in a single patient. As the study did not specifically address frequency differences, we reached out to the first author of the study, Dr. Lynette Sadleir. She provided two chapters from her PhD thesis wherein she describes slight frequency differences in spontaneous versus hyperventilation-induced seizures: seizures induced by hyperventilation are slower. That said, in our email exchange, Dr. Sadleir mentions that:

“In a nut shell we found that absence seizures in HV do have some differences both electrographically and clinically as you can see in our analysis. Having said that, they are more similar electrographically than they are different.”

Our current study in rats supports this statement.

The conclusion that spontaneous and hyperventilation-induced seizures are generally similar further inform our model regarding respiratory alkalosis and spike-wave seizures. We speculate that spontaneous and hyperventilation-provoked spike-wave seizures share similar neural circuits. Thus, we hypothesize that the primary role played by respiratory alkalosis is to provoke an existing circuit that also generates spontaneous spike-wave seizures; once set in motion, either spontaneously or by alkalosis, the circuit simply executes a series of events that result in a generally, highly stereotyped spike-wave seizure. We are quite excited at this possibility as it suggests that resolving how alkalosis reliably and robustly provokes seizures will inform mechanisms that are also relevant for spontaneous spike-wave seizures.

We also agree that including post-hoc confirmation of RTN targeting is important. We verified expression of channelrhodopsin and fiber placement in the RTN after each experiment. Regrettably, however, we did not include these data in the original manuscript. We agree, most readers would appreciate post-hoc confirmation and we have now included a representative image in Figure 5B. Thanks for the suggestion.

2) The abstract should contain more information about the experiments utilizing channelrhodopsin to manipulate neural activity, and how this information was interpreted. As of now, the data could missed by many readers.

Thanks for this suggestion – we now include this information in the abstract.

3) Readers will want some clarification to interpret the correlation between pH and electrical activity of intralaminar neurons.

We now include this clarification in the discussion (see Lines 406). Moreover, the inclusion of new electrophysiological experiments that more directly evaluate the correlation between pH and intralaminar activity will likely make this point more clearly.

4) More insight could be offered into the circuit mechanisms by which intralaminar neurons initiate or predispose circuits to generating spike-wave seizures.

We now include this clarification in the discussion (see Line 363).

5) Addressing the issue of whether intralaminar thalamus neurons are pH sensitive with further controls.

We agree with this comment. We now include supportive data derived from electrophysiological recordings of individual intralaminar neurons (Figure 6F-H). Consistent with our calcium imaging results, patch-clamp recordings during pH manipulations demonstrate that extracellular alkalosis generates inward (i.e., depolarizing) currents in intralaminar neurons. These inward currents are associated with a decreased membrane resistance, suggesting that alkalosis opens ion channels expressed by intralaminar neurons. Indeed, we have preliminary data that further indicates that alkalosis largely activates glutamatergic currents in intralaminar neurons (see Author response image 1). We are not keen to include these data in the current manuscript because (1) they remain preliminary, and (2) require substantial investigation to identify their source. We do, however, include data showing that intralaminar neurons appear more pH-sensitive than other proposed spike-wave seizure-generating nodes (i.e., somatosensory cortex and somatosensory thalamus, see Figure 6H). We believe these data further support the hypothesis that intralaminar neurons play an important role in the generation of alkalosis-provoked (and perhaps spontaneous) spike-wave seizures. We include some of these points in the discussion (see Line 406). Regrettably, we acknowledge that our sample size for these new patch clamp recordings is relatively low. Unfortunately, such recordings are difficult to obtain as many pups from WAG/Rij litters are hydrocephalic and are not viable. We can perform more recordings, but are concerned with how much more time we will need to generate more data for these panels – we hope the reviewers understand.

**Author response image 1. sa2fig1:** Glutamate receptor blockade reduces alkalization-induced inward currents evoked in intralaminar neurons. A. Representative voltage-clamp recordings of intralaminar neuron during bath application of control (pH 7.3) and alkaline (pH 8.0) conditions. The gray symbols represent a control neuron. Green symbols represent a neuron undergoing pH manipulations during application of kynurenic acid, a glutamate receptor blocker. The red symbols represent a neuron undergoing pH manipulations during application of TTX. B. Population response. Control, n = 4. Kynurenic acid, n = 4. TTX, n = 5.

Reviewer #1:– The conclusions of the optogenetic stimulation of RTN would be strengthened by showing post-hoc validation of the viral targeting in a spatially specific manner. Can the authors confirm that those experiments accurately targeted the RTN and not other regions more broadly?

Thank you for the suggestion. We have now included a representative image of channelrhodopsin expression in the RTN of an experimental animal (Figure 5B).

– Do the manipulations (decreased O2, decreased 02/elevated C02, elevated C02 in normoxia, optogenetic stimulation of RTN) alter the duration or frequency components of SWSs? Or are the seizures identical across conditions, but their incidence is changed?

This is a great question, as it gets at whether SWSs provoked by hyperventilation are similar to those occurring spontaneously. We have now included these measures in the revised manuscript. By and large, seizure duration is identical across conditions. In a few cases, spike-wave seizures were slower during the manipulations. These findings are summarized in Tables 2 and 3. It remains too early to understand what mechanisms drive any differences with spike-wave seizure properties. That said, parallels can be found with human spike-wave seizures (Sadleir et al., 2008, Epilepsia 49(12):2100-7). Thanks for suggesting that we look into it.

– The alignment of the pH manipulations and changes in GCaMP DF/F in Figure 6D are not clear. It appears that changes in DF/F are not time locked to changes in pH? Is that correct? If so what accounts for this time lag?

Correct, the change in DF/F lags behind the indicated solution change. This lag is also evident in our patch clamp recordings. The duration required for the new solution to fill the perfusion tubes and the recording chamber generally takes 1-2 minutes in our experiments and accounts for the lag. We now address this apparent delayed response in the legend of figure 6.

– Why is the change in respiration rate during optogenetic stimulation of the intralaminar thalamus so variable when stimulated during normoxia (Figure 5E) but so robust and uniform when done in high C02 (Figure 5H). Is this because the relationship between breathing rate the pH is different in different conditions or is it due to variability in targeting of the RTN?

The effects of CO_2_ and hypoxia on respiration are always very reliable and robust. The reviewer is correct to point out that optogenetically-induced hyperventilation is more variable. (Although we assume the reviewer is referring to optogenetic stimulation of the retrotrapezoid nucleus, not the intralaminar nucleus; we did not optogenetically manipulate the latter structure). We hypothesize that the variability partly reflects differences in channelrhodopsin expression and/or fiber optic placement. Thus, while all animals respond to laser stimulation with hyperventilation, we speculate that the more robust responders represent animals with high opsin expression and ideal fiber optic placement; as the RTN is a very small, deep structure, fiber optic placement can be tricky. That said, post-hoc analyses confirmed that all animals had sufficient opsin expression and fiber placement. However, in the end, it’s difficult to know if the number of laser-activated RTN neurons was equivalent across all animals. We suspect that this number was somewhat variable across animals and, therefore, results in a variable hyperventilation response.

– A better description of the cFos thresholding quantification should be included. It is not clear what was done there. A clearer description (visually in the figure and/or in the text) of the statistical comparisons made for each condition/threshold should also be included. This could be added as a supplemental figure if it make the presentation easier.

Yes, regrettably, we did not include a better description of our thresholding procedure. We now include a clearer description in the Methods (Line 571).

– The discussion would be strengthened by considering how increased input from intralaminar thalamic neurons trigger SWS. The authors nicely consider how intralaminar thalamic neuron activity may arise from increased excitatory input, but there is not much consideration of how increased output from intralaminar thalamic neurons feed into canonical SWS circuitry (nRT, VB, cortex).

Great question – we’ve thought quite a bit about this. In the end, we currently don’t know but experiments are ongoing to attempt this very question. We are happy to speculate in the discussion (Line 363).

Reviewer #2:– It would be helpful to see viral expression and cannula placement to determine if hyperventilation and SWS induction was truly due to activation of RTN neurons. The C1 region, amongst other medullary regions, are in close proximity to the RTN.

Thanks for the suggestion. We performed post-hoc analyses of opsin expression and fiber placement. We now include a representative image in Figure 5B. The reviewer is correct to point out that our virus, using the transcription factor Phox2b, does indeed target C1 and RTN neurons. It is possible to differentiate these populations using transgenic approaches (see Souza et al., 2020. J.Neuro.), however, we do not have a current seizure model bred into a Th-Cre background. We agree that future studies using the Gria4 and Th-Cre mice would allow us to differentiate the contribution of each population to hyperventilation-induced SWSs. As the reviewer suggests, these are important experiments to pursue.

– There seems to be a disconnect in SWS generation and the level of hyperventilation reached – see 5D and 5E – some of the animals with the highest increase in SWS had a minimal change in respiratory rate. Do individual animals have a varying pH change related to hyperventilation (perhaps looking at respiratory rate change and pH and arterial CO2 in the animals from figure 3 would be illustrative) or is there another explanation that could be suggested?

The reviewer brings up a fair point. We suspect that the apparent disconnect largely reflects the high variability in SWS expression in these animals. Indeed, quantifying spontaneous seizures in these animals is vexing. By contrast, the blood measurements are less variable and therefore likely don’t explain the variable seizure response. Thus, we speculate that overlaid on top of our hyperventilation manipulations is a generally highly variable seizure occurrence that can sometimes result in an apparent mismatch between seizure count and respiration rate.

By way of a silly analogy, imagine one is tasked to pick up marbles from a cup while sitting in the back seat of a car driving on a highway at night. In the control condition, you must do so in the dark. In the experimental condition, you can use a flashlight. Most individuals will pick up more marbles in the experimental condition, thereby supporting the hypothesis that light helps individuals pick up marbles.

Now imagine performing the same experiment on a dirt road wherein unpredictable bumps make the marble picking task more difficult. In some cases, we suspect that the features of the bumpy road will override the utility of a flashlight, causing an individual to sometimes pick fewer marbles in the light.

While silly, this is how we think about the apparent disconnects between SWS generation and hyperventilation. Sometimes the vexing and unpredictable spontaneous seizure occurrence will override the effects of hyperventilation.

– It's stated that only animals that responded to optical stimulation with an increase in respiratory frequency were included in the analysis shown in Figure 5. Did you determine a reason why some animals did not respond – such as poor viral targeting, expression or poor cannula targeting? it would be illuminative to provide the numbers for those that had accurate targeting but no response to optical stimulation. Also, it appears that one animal that had a decrease in respiratory frequency (the dark purple Figure 5E) was included in the analysis despite the exclusion criteria listed.

Yes, this is true. While only one of our experimental animals did not consistently respond to laser stimulation with hyperventilation, this animal did so because of virus injection/fiber placement. We included the purple animal in our study because the animal did demonstrate a moderate response to optogenetic activation during pre-experimental testing. That said, the reviewer brings up a fair point about the purple animal. The apparent decrease in rate likely reflects our approach to quantifying respiration in Figure 5G (old 5E): we simply calculated the mean frequency per bin before and during stimulation. We feel that this is the least complicated, most objective way to quantify rates.

As shown in the binned respiration rates below for a subset of animals in 5E, optogenetic stimulation increased rate for most animals (Author response image 2) . However, the purple animal in question exhibited a transient increase in rate from 12 to 4 minutes prior to optogenetic simulation, thereby driving a higher, pre-stimulation mean. We suspect that this transient increase contributes to the apparent disconnect in the purple animal. If the reviewer feels that we should omit this animal, then we are happy to do so.

**Author response image 2. sa2fig2:** Respiratory measurements in a subset of animals shown in Figure 5G. In all but one animal (purple), optogenetic stimulation clearly increased respiration rate. Unfortunately, an unsteady baseline rate can explain the apparent decrease in rate in the purple animal.

– Did you quantify cfos activation in other thalamic regions of interest such as the reticular nucleus, was the intralaminar region the only region with significant activation?

We primarily focused our attention on those regions enriched cFos expression. The reviewer’s astute comment is warranted and may be motivated by the role the thalamic reticular nucleus plays in SWS generation. We did not see much cFos enrichment in the reticular nucleus, an observation we suspect is the result of first injecting the animals with ethosuximide to inhibit seizure production. While we describe this experimental strategy in the main text, we now include it again in the legend of the figure.

– As for the calcium imaging, the sample imaging does not provide enough anatomical context for orientation. Also, as alkaline pH can be a general neuronal activator and we might expect results similar to what was obtained looking at a number of brain regions. To make the case that the intralaminar nucleus is particularly or selectively activated requires a control comparison. Perhaps the VB thalamus as it is relevant to the potential mechanisms and anatomically adjacent.

The reviewer is correct to point out the limitations of the data we originally presented. We now include additional data using electrophysiology to support the hypothesis that intralaminar neurons are pH sensitive. Consistent with our calcium imaging results, our patch clamp data demonstrate that alkaline conditions evoke inward currents in intralaminar neurons (Figure 6F-G). Comparable recordings and manipulations in somatosensory cortex (S1) and VB thalamus show that these other SWS circuit nodes are less sensitive to alkaline conditions (Figure 6H). Indeed, these data support an older report wherein VB neuron sensitivity to alkaline conditions was measured (Meuth et al., 2006, J Physiol). Interestingly, Meuth et al., show that VB neurons express ion channels that are pH sensitive, but the aggregate response to alkalization is minimal as the ion channel activities cancel each other out. In fact, the authors conclude that VB thalamus is quite resistant to pH changes. We now highlight this work in the Discussion.

– Was cfos activation driven by RTN opto stimulation evaluated?

Regrettably, we did not look. Instead, we have opted for more direct measurements of neuronal activation, but this work is ongoing. In brief, we have recently translated our model of hyperventilation-provoked SWS to mouse model (Gria4). As in our rats, hyperventilation provokes a robust increase in SWSs in the Gria4 mouse. With this model, we now have many more transgenic tools at our disposal. For example, we have crossed the Gria4 mouse with a GCaMP-expressing mouse so that we can measure activity changes in real time. We are also performing silicone probe recordings of several thalamic regions so that we can measure activity in response to hypoxia and optogenetic hyperventilation. These data remain preliminary. But yes, we wholeheartedly agree with the premise of the reviewer’s question. Upon finalizing this project, however, we opted to translate our model to a more tractable system and hope to address this and other questions in our follow up study.

The discussion nicely discusses the current extant literature and the context of this study as well as the limitations of this manuscript.It would enhance the impact of the study if some of these points were addressed experimentally, as currently while the manuscript very elegantly and nicely supports that respiratory alkalosis from hyperventilation drives seizure induction, the critical nature and the role of the intralaminar thalamus is less supported – is it truly necessary for the generation of spike wave discharges. Also whether it has a direct or more indirect – potentiated by other inputs – role in chemosensation is not clear.– Patch-clamp electrophysiology of intralaminar neurons could readily determine if there is any intrinsic pH sensitivity of these neurons, looking for a change in firing due to changes in pH. Answering the question of whether this is presynaptically (maybe from reticular nuclei input) could also be addressed by looking at changes in spontaneous transmission.

We have now included patch clamp data from intralaminar neurons, as well as somatosensory and VB neurons. Our included data show that intralaminar neurons appear to have heightened pH sensitivity, relative to cortical and VB neurons. We also have preliminary data that addresses the reviewer’s comment regarding changes in synaptic excitability. In brief, our data suggest that alkaline-mediated activation of intralaminar neurons involves the activation of glutamatergic inputs. The robust, alkaline-evoked inward currents in intralaminar neurons are significantly attenuated when bathing the slice in kynurenic acid (Rebuttal Figure 2). We are keenly interested in resolving the source of such excitation and are currently pursuing leads. That said, we feel that a comprehensive study of such inputs is likely beyond the scope of the current study. We hope that the reviewer understands that we aim to present such data in a follow-up study.

– Following electrophysiologic studies up with evaluation of expression of pH sensitive ion channels/receptors would be important as noted in your discussion.

We completely agree and are currently performing experiments to resolve this important point.

– It would be interesting to cfos expression in response to respiratory alkalosis in reticular regions and the dorsal raphe – which may be more chemosensitive and project to the intralaminar nucleus.

We 100% agree. We are keenly interested in this point and hope that utilizing the clear advantage of mouse genetics, we have positioned ourselves to address this and other questions using more refined techniques. Our primary excitement of the current manuscript is that establishes the basic features necessary to evoke SWS with hyperventilation using a well-established model of absence epilepsy. That said, in moving forward, we feel that addressing the important mechanistic questions is best suited in the mouse. In short, we completely agree with all of these reviewer’s comments, and we appreciate their interest. We feel that we have possibly opened the door to some very interesting follow up experiments – and now we can begin to address these and other important questions with follow up studies.

– While this may be beyond the scope of the manuscript, optogenetic activation of the intralaminar nucleus to produce spike wave discharges and seizures, and also doing this in the context of hypoxia-induced hyperventilation or rescue with excessive CO2 would be very powerful and more fully evaluate the model in Figure 7.

We agree completely. We are pursuing this and other experiments in our Gria4 mouse model. We hope that the reviewer understands that while we 100% agree with the comment, we are still in the midst of addressing this and other questions in a much more tractable mouse system.